# Generalizing and Decoupling Neural Collapse via Hyperspherical Uniformity Gap

**Weiyang Liu[1,2,*], Longhui Yu[3,*], Adrian Weller[2,4], Bernhard Schölkopf[1]**

[1]Max Planck Institute for Intelligent Systems - Tübingen    [2]University of Cambridge
[3]Peking University    [4]The Alan Turing Institute

## Abstract

The neural collapse (NC) phenomenon describes an underlying geometric symmetry for deep neural networks, where both deeply learned features and classifiers converge to a simplex equiangular tight frame. It has been shown that both cross-entropy loss and mean square error can provably lead to NC. We remove NC's key assumption on the feature dimension and the number of classes, and then present a generalized neural collapse (GNC) hypothesis that effectively subsumes the original NC. Inspired by how NC characterizes the training target of neural networks, we decouple GNC into two objectives: minimal intra-class variability and maximal inter-class separability. We then use hyperspherical uniformity (which characterizes the degree of uniformity on the unit hypersphere) as a unified framework to quantify these two objectives. Finally, we propose a general objective – hyperspherical uniformity gap (HUG), which is defined by the difference between inter-class and intra-class hyperspherical uniformity. HUG not only provably converges to GNC, but also decouples GNC into two separate objectives. Unlike cross-entropy loss that couples intra-class compactness and inter-class separability, HUG enjoys more flexibility and serves as a good alternative loss function. Empirical results show that HUG works well in terms of generalization and robustness.

## 1 Introduction

Recent years have witnessed the great success of deep representation learning in a variety of applications ranging from computer vision [37], natural language processing [16] to game playing [55, 64]. Despite such a success, how deep representations can generalize to unseen scenarios and when they might fail remain a black box. Deep representations are typically learned by a multi-layer network with cross-entropy (CE) loss optimized by stochastic gradient descent. In this simple setup, [86] has shown that zero loss can be achieved even with arbitrary label assignment. After continuing to train the neural network past zero loss with CE, [60] discovers an intriguing phenomenon called neural collapse (NC). NC can be summarized as the following characteristics:

- **Intra-class variability collapse**: Intra-class variability of last-layer features collapses to zero, indicating that all the features of the same class concentrate to their intra-class feature mean.
- **Convergence to simplex ETF**: After being centered at their global mean, the class-means are both linearly separable and maximally distant on a hypersphere. Formally, the class-means form a simplex equiangular tight frame (ETF) which is a symmetric structure defined by a set of maximally distant and pair-wise equiangular points on a hypersphere.
- **Convergence to self-duality**: The linear classifiers, which live in the dual vector space to that of the class-means, converge to their corresponding class-mean and also form a simplex ETF.
- **Nearest decision rule**: The linear classifiers behave like nearest class-mean classifiers.

The NC phenomenon suggests two general principles for deeply learned features and classifiers: minimal intra-class compactness of features (*i.e.*, features of the same class collapse to a single point), and maximal inter-class separability of classifiers / feature mean (*i.e.*, classifiers of different classes have maximal angular margins). While these two principles are largely independent, popular loss functions such as CE and square error (MSE) completely couple these two principles together. Since there is no trivial way for CE and MSE to decouple these two principles, we identify a novel quantity – hyperspherical uniformity gap (HUG), which not only characterizes intra-class feature compactness and inter-class classifier separability as a whole, but also fully decouples these two

principles. The decoupling enables HUG to separately model intra-class compactness and inter-class separability, making it highly flexible. More importantly, HUG can be directly optimized and used to train neural networks, serving as an alternative loss function in place of CE and MSE for classification. HUG is formulated as the difference between inter-class and intra-class hyperspherical uniformity. Hyperspherical uniformity [48] quantifies the uniformity of a set of vectors on a hypersphere and is used to capture how diverse these vectors are on a hypersphere. Thanks to the flexibility of HUG, we are able to use many different formulations to characterize hyperspherical uniformity, including (but not limited to) minimum hyperspherical energy (MHE) [45], maximum hyperspherical separation (MHS) [48] and maximum gram determinant (MGD) [48]. Different formulations yield different interpretation and optimization difficulty (*e.g.*, HUG with MHE is easy to optimize, HUG with MGD has interesting connection to geometric volume), thus leading to different performance.

Similar to CE loss, HUG also provably leads to NC under the setting of unconstrained features [53]. Going beyond NC, we hypothesize a generalized NC (GNC) with hyperspherical uniformity, which extends the original NC to the scenario where there is no constraint for the number of classes and the feature dimension. NC requires the feature dimension no smaller than the number of classes while GNC no longer requires this. We further prove that HUG also leads to GNC at its objective minimum.

Another motivation behind HUG comes from the classic Fisher discriminant analysis (FDA) [19] where the basic idea is to find a projection matrix $\boldsymbol{T}$ that maximizes between-class variance and minimizes within-class variance. What if we directly optimize the input data (without any projection) rather than optimizing the linear projection in FDA? We make a simple derivation below:

$$\textbf{Projection FDA:} \quad \max_{\boldsymbol{T} \in \mathbb{R}^{d \times r}} \operatorname{tr}\left( \left( \boldsymbol{T}^\top \boldsymbol{S}_w \boldsymbol{T} \right)^{-1} \boldsymbol{T}^\top \boldsymbol{S}_b \boldsymbol{T} \right) \qquad \textbf{Data FDA:} \quad \max_{\boldsymbol{x}_1, \cdots, \boldsymbol{x}_n \in \mathbb{S}^{d-1}} \operatorname{tr}\left( \boldsymbol{S}_b \right) - \operatorname{tr}\left( \boldsymbol{S}_w \right)$$

where the between-class scatter matrix is $\boldsymbol{S}_w = \sum_{i=1}^{C} \sum_{j \in A_c} (\boldsymbol{x}_j - \boldsymbol{\mu}_i)(\boldsymbol{x}_j - \boldsymbol{\mu}_i)^\top$, the within-class scatter matrix is $\boldsymbol{S}_b = \sum_{i=1}^{C} n_i (\boldsymbol{\mu}_i - \bar{\boldsymbol{\mu}})(\boldsymbol{\mu}_i - \bar{\boldsymbol{\mu}})^\top$, $n_i$ is the number of samples in the $i$-th class, $n$ is the total number of samples, $\boldsymbol{\mu}_i = n_i^{-1} \sum_{j \in A_c} \boldsymbol{x}_j$ is the $i$-th class-mean, and $\bar{\boldsymbol{\mu}} = n^{-1} \sum_{j=1}^{n} \boldsymbol{x}_j$ is the global mean. By considering class-balanced data on the unit hypersphere, optimizing data FDA is equivalent to simultaneously maximizing $\operatorname{tr}(\boldsymbol{S}_b)$ and minimizing $\operatorname{tr}(\boldsymbol{S}_w)$. Maximizing $\operatorname{tr}(\boldsymbol{S}_b)$ encourages inter-class separability and is a necessary condition for hyperspherical uniformity.[1] Minimizing $\operatorname{tr}(\boldsymbol{S}_w)$ encourages intra-class feature collapse, reducing intra-class variability. Therefore, HUG can be viewed a generalized FDA criterion for learning maximally discriminative features.

However, one may ask the following questions: *Why is HUG useful if we already have the FDA criterion? Could we simply optimize data FDA?* In fact, the FDA criterion has many degenerate solutions. For example, we consider a scenario of 10-class balanced data where all features from the first 5 classes collapse to the north pole on the unit hypersphere and features from the rest 5 classes collapse to the south pole on the unit hypersphere. In this case, $\operatorname{tr}(\boldsymbol{S}_w)$ is already minimized since it achieves the minimum zero. $\operatorname{tr}(\boldsymbol{S}_b)$ also achieves its maximum $n$ at the same time. In contrast, HUG naturally generalizes FDA without having these degenerate solutions and serves as a more reliable criterion for training neural networks. We summarize our contributions below:

- We decouple the NC phenomenon into two separate learning objectives: maximal inter-class separability (*i.e.*, maximally distant class feature mean and classifiers on the hypersphere) and minimal intra-class variability (*i.e.*, intra-class features collapse to a single point on the hypersphere).

- Based on the two principled objectives induced by NC, we hypothesize the generalized NC which generalizes NC by dropping the constraint on the feature dimension and the number of classes.

- We identify a general quantity called hyperspherical uniformity gap, which well characterizes both inter-class separability and intra-class variability. Different from the widely used CE loss, HUG naturally decouples both principles and thus enjoys better modeling flexibility.

- Under the HUG framework, we consider three different choices for characterizing hyperspherical uniformity: minimum hyperspherical energy, maximum hyperspherical separation and maximum Gram determinant. HUG provides a unified framework for using different characterizations of hyperspherical uniformity to design new loss functions.

---

[1]We first obtain the upper bound $n$ of $\operatorname{tr}(\boldsymbol{S}_b)$ from $\operatorname{tr}(\boldsymbol{S}_b) = \sum_{i=1}^{C} n_i \|\boldsymbol{\mu}_i - \bar{\boldsymbol{\mu}}\|_F^2 \leq \sum_{i=1}^{C} n_i \|\boldsymbol{\mu}_i\| \cdot \|\bar{\boldsymbol{\mu}}\| \leq n$. Because a set of vectors $\{\boldsymbol{\mu}_i\}_{i=1}^{n}$ achieving hyperspherical uniformity has $\mathbb{E}_{\boldsymbol{\mu}_1, \cdots, \boldsymbol{\mu}_n}\{\|\bar{\boldsymbol{\mu}}\|\} \to \boldsymbol{0}$ (as $n$ grows larger) [20]. Then we have that $\operatorname{tr}(\boldsymbol{S}_b)$ attains $n$. Therefore, vectors achieving hyperspherical uniformity are one of its maximizers. $\operatorname{tr}(\boldsymbol{S}_w)$ can simultaneously attain its minimum if intra-class features collapse to a single point.

## 2   ON GENERALIZING AND DECOUPLING NEURAL COLLAPSE

NC describes an intriguing phenomenon for the distribution of last-layer features and classifiers in overly-trained neural networks, where both features and classifiers converge to ETF. However, ETF can only exist when the feature dimension $d$ and the number of classes $C$ satisfy $d \geq C - 1$. This is not always true for deep neural networks. For example, neural networks for face recognition are usually trained by classifying large number of classes (*e.g.*, more than 85K classes in [23]), and the feature dimension (*e.g.*, 512 in SphereFace [43]) is usually much smaller than the number of classes. In general, when the number of classes is already large, it is prohibitive to use a larger feature dimension. Thus a question arises: *what will happen in this case if a neural network is fully trained?*

Motivated by this question, we conduct a simple experiment to simulate the case of $d \geq C - 1$ and the case of $d < C - 1$. Specifically, we train a convolutional neural network (CNN) on MNIST with feature dimension 2. For the case of $d \geq C - 1$, we use only 3 classes (digit 0,1,2) as the training set. For the case of $d < C - 1$, we use all 10 classes as the training set. We visualize the learned features of both cases in Figure 1. The results verify the case of $d \geq C - 1$ indeed approaches to NC, and ETF does not exist in the case of $d < C - 1$. Inter-

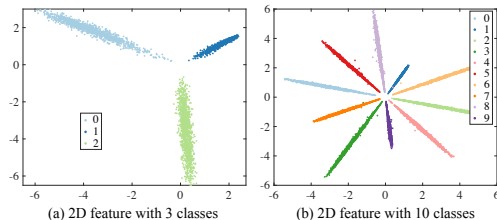

Figure 1: 2D learned feature visualization on MNIST. The features are inherently 2-dimensional and are plotted without visualization tools. (a) Case 1: $d = 2, C = 3$; (b) Case 2: $d = 2, C = 10$.

estingly, one can observe that learned features in both cases approach to the configuration of equally spaced frames on the hypersphere. To accommodate the case of $d < C - 1$, we extend NC to the generalized NC by hypothesizing that last-layer inter-class features and classifiers converge to equally spaced points on the hypersphere, which can be characterized by hyperspherical uniformity.

---

**Generalized Neural Collapse (GNC)**

We define the feature global mean as $\boldsymbol{\mu}_G = \text{Ave}_{i,c} \boldsymbol{x}_{i,c}$ where $\boldsymbol{x}_{i,c} \in \mathbb{R}^d$ is the last-layer feature of the $i$-th sample in the $c$-th class, the feature class-mean as $\boldsymbol{\mu}_c = \text{Ave}_i \boldsymbol{x}_{i,c}$ for different classes $c \in \{1, \cdots, C\}$, the feature within-class covariance as $\boldsymbol{\Sigma}_W = \text{Ave}_{i,c}(\boldsymbol{x}_{i,c} - \boldsymbol{\mu}_c)(\boldsymbol{x}_{i,c} - \boldsymbol{\mu}_c)^\top$ and the feature between-class covariance as $\boldsymbol{\Sigma}_B = \text{Ave}_c(\boldsymbol{\mu}_c - \boldsymbol{\mu}_G)(\boldsymbol{\mu}_c - \boldsymbol{\mu}_G)^\top$. GNC states that

- **(1) Intra-class variability collapse**: Intra-class variability of last-layer features collapse to zero, indicating that all the features of the same class converge to their intra-class feature mean. Formally, GNC has that $\boldsymbol{\Sigma}_B^\dagger \boldsymbol{\Sigma}_W \to \mathbf{0}$ where $\dagger$ denotes the Moore-Penrose pseudoinverse.

- **(2) Convergence to hyperspherical uniformity**: After being centered at their global mean, the class-means are both linearly separable and maximally distant on a hypersphere. Formally, the class-means converge to equally spaced points on a hypersphere, *i.e.*,

$$\sum_{c \neq c'} K(\hat{\boldsymbol{\mu}}_c, \hat{\boldsymbol{\mu}}_{c'}) \to \min_{\hat{\boldsymbol{\mu}}_1, \cdots, \hat{\boldsymbol{\mu}}_C} \sum_{c \neq c'} K(\hat{\boldsymbol{\mu}}_c, \hat{\boldsymbol{\mu}}_{c'}), \quad \|\boldsymbol{\mu}_c - \boldsymbol{\mu}_G\| - \|\boldsymbol{\mu}_{c'} - \boldsymbol{\mu}_G\| \to 0, \ \forall c \neq c' \quad (1)$$

  where $\hat{\boldsymbol{\mu}}_i = \|\boldsymbol{\mu}_i - \boldsymbol{\mu}_G\|^{-1}(\boldsymbol{\mu}_i - \boldsymbol{\mu}_G)$ and $K(\cdot, \cdot)$ is a kernel function that models pairwise interaction. Typically, we consider Riesz $s$-kernel $K_s(\hat{\boldsymbol{\mu}}_c, \hat{\boldsymbol{\mu}}_{c'}) = \text{sign}(s) \cdot \|\hat{\boldsymbol{\mu}}_c - \hat{\boldsymbol{\mu}}_{c'}\|^{-s}$ or logarithmic kernel $K_{\log}(\hat{\boldsymbol{\mu}}_c, \hat{\boldsymbol{\mu}}_{c'}) = \log \|\hat{\boldsymbol{\mu}}_c - \hat{\boldsymbol{\mu}}_{c'}\|^{-1}$. For example, the Riesz $s$-kernel with $s = d - 2$ is a variational characterization of hyperspherical uniformity (*e.g.*, hyperspherical energy [45]) using Newtonian potentials. In the case of $d = 3, s = 1$, the Riesz kernel is called Coulomb potential and the problem of finding minimal coulomb energy is called Thomson problem [70].

- **(3) Convergence to self-duality**: The linear classifiers, which live in the dual vector space to that of the class-means, converge to their corresponding class-means, leading to hyperspherical uniformity. Formally, GNC has that $\|\boldsymbol{w}_c\|^{-1}\boldsymbol{w}_c - \hat{\boldsymbol{\mu}}_c \to 0$ where $\boldsymbol{w}_c \in \mathbb{R}^d$ is the $c$-th classifier.

- **(4) Nearest decision rule**: The learned linear classifiers behave like the nearest class-mean classifiers. Formally, GNC has that $\arg\max_c \langle \boldsymbol{w}_c, \boldsymbol{x} \rangle + b_c \to \arg\min_c \|\boldsymbol{x} - \boldsymbol{\mu}_c\|$.

---

In contrast to NC, GNC further considers the case of $d < C - 1$ and hypothesizes that both feature class-means and classifiers converge to hyperspherically uniform point configuration that minimizes some form of pairwise potentials. Similar to how NC connects tight frame theory [74] to deep learning, our GNC hypothesis connects potential theory [3] to deep learning, which may shed new light on understanding it. We show in Theorem 1 that GNC reduces to NC in the case of $d \geq C - 1$.

**Theorem 1 (Regular Simplex Optimum for GNC)** *Let $f : (0, 4] \to \mathbb{R}$ be a convex and decreasing function defined at $v = 0$ by $\lim_{v \to 0^+} f(v)$. If $2 \leq C \leq d+1$, then we have that the vertices of regular $(C-1)$-simplices inscribed in $\mathbb{S}^{d-1}$ with centers at the origin (equivalent to simplex ETF) minimize the hyperspherical energy $\sum_{c \neq c'} K(\hat{\boldsymbol{\mu}}_c, \hat{\boldsymbol{\mu}}_{c'})$ on the unit hypersphere $\mathbb{S}^{d-1}$ ($d \geq 3$) with the kernel as $K(\hat{\boldsymbol{\mu}}_c, \hat{\boldsymbol{\mu}}_{c'}) = f(\|\hat{\boldsymbol{\mu}}_c - \hat{\boldsymbol{\mu}}_{c'}\|^2)$. If $f$ is strictly convex and strictly decreasing, then these are the only energy minimizing $C$-point configurations. Thus GNC reduces to NC when $d \geq C-1$.*

We note that Theorem 2 guarantees the simplex ETF as the minimizer of a general family of hyperspherical energies (as long as $f$ is convex and decreasing). This suggests that there are many possible kernel functions $K(\cdot, \cdot)$ in GNC that can effectively generalize NC. The case of $d < C-1$ is where GNC really gets interesting but complicated. Other than the regular simplex case, we also highlight a special uniformity case of $2d = C$. In this case, we can prove in Theorem 2 that GNC(2) converges to the vertices of a cross-polytope as hyperspherical energy gets minimized. As the number of classes gets infinitely large, we show in Theorem 3 that GNC(2) leads to a point configuration that is uniformly distributed

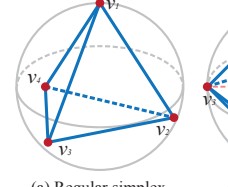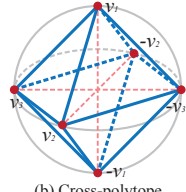

(a) Regular simplex     (b) Cross-polytope

Figure 2: Geometric illustration in $\mathbb{R}^3$ of (a) regular simplex optimum (equivalent to simplex ETF in NC) and (b) cross-polytope optimum in GNC.

on $\mathbb{S}^{d-1}$. Additionally, we show a simple yet interesting result in Proposition 1 that the last-layer classifiers are already initialized to be uniformly distributed on the hypersphere in practice.

**Theorem 2 (Cross-polytope Optimum for GNC)** *If $C = 2d$, then the vertices of the cross-polytope are the minimizer of the hyperspherical energy in GNC(2).*

The cross-polytope optimum for GNC(2) is in fact quite intuitive, because it corresponds to the Cartesian coordinate system (up to a rotation). For example, the vertices of the unit cross-polytope in $\mathbb{R}^3$ are $(\pm 1, 0, 0), (0, \pm 1, 0), (0, 0, \pm 1)$. These 6 vectors minimize the hyperspherical energy on $\mathbb{S}^2$. We illustrate both the regular simplex and cross-polytope cases in Figure 2. For the other cases of $d < C-1$, there exists generally no simple and universal point structure that minimizes the hyperspherical energy, as heavily studied in [12, 27, 38, 63]. For the point configurations that asymptotically minimize the hyperspherical energy as $C$ grows larger, Theorem 3 can guarantee that these configurations asymptotically converge to a uniform distribution on the hypersphere.

**Theorem 3 (Asymptotic Convergence to Hyperspherical Uniformity)** *Consider a sequence of point configurations $\{\hat{\boldsymbol{\mu}}_1^C, \cdots, \hat{\boldsymbol{\mu}}_C^C\}_{C=2}^\infty$ that asymptotically minimizes the hyperspherical energy on $\mathbb{S}^{d-1}$ as $C \to \infty$, then $\{\hat{\boldsymbol{\mu}}_1^C, \cdots, \hat{\boldsymbol{\mu}}_C^C\}_{C=2}^\infty$ is uniformly distributed on the hypersphere $\mathbb{S}^{d-1}$.*

**Proposition 1 (Minimum Energy Initialization)** *With zero-mean Gaussian initialization (e.g., [22, 28]), the $C$ last-layer classifiers of neural networks are initialized as a uniform distribution on the hypersphere. The expected initial energy is $C(C-1) \int_{\mathbb{S}^{d-1}} \int_{\mathbb{S}^{d-1}} \|\hat{\boldsymbol{\mu}}_c - \hat{\boldsymbol{\mu}}_{c'}\|^{-2} d\sigma_{d-1}(\hat{\boldsymbol{\mu}}_c) d\sigma_{d-1}(\hat{\boldsymbol{\mu}}_{c'})$.*

With Proposition 1, one can expect that the hyperspherical energy of the last-layer classifiers will first increase and then decrease to a lower value than the initial energy. To validate the effectiveness of our GNC hypothesis, we conduct a few experiments to show how both class feature means and classifiers converge to hyperspherical uniformity (*i.e.*, minimizing the hyperspherical energy), and how intra-class feature variability collapses to almost zero. We start with an intuitive understanding about GNC from Figure 1. The results are directly produced by the learned features without any visualization tool (such as t-SNE [73]), so the feature distribution can reflect the underlying one learned by neural networks. We observe that GNC is attained in both $d < C-1$ and $d \geq C-1$, while NC is violated in $d < C-1$ since the learned feature class-means can no longer form a simplex ETF. To see whether the same conclusion holds for higher feature dimensions, we also train two CNNs on CIFAR-100 with feature dimension as 64 and 128, respectively. The results are given in Figure 3.

Figure 3 shows that GNC captures well the underlying convergence of the neural network training. Figure 3(a,c) shows that the hyperspherical energy of feature class-means and classifiers converge to a small value, verifying the correctness of GNC(2) and GNC(3) which indicate both feature class-means and classifiers converge to hyperspherical uniformity. More interestingly, in the MNIST experiment, we can compute the exact minimal energy on $\mathbb{S}^1$: 2 in the case of $d=2, C=3$ (1/3 for average energy) and $\approx 82.5$ in the case of $d=2, C=10$ ($\approx 0.917$ for average energy). The final average energy in Figure 3(a) matches our theoretical minimum well. From Figure 3(c), we observe that the classifier energy stays close to its minimum at the very beginning, which matches our Proposition 1

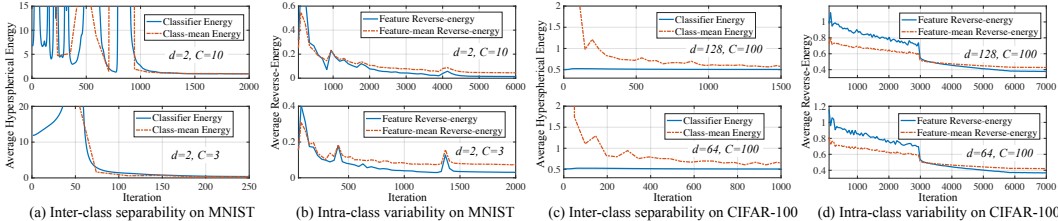

Figure 3: Training dynamics of hyperspherical energy (which captures inter-class separability) and hyperspherical reverse-energy (which captures intra-class variability). (a,b) MNIST with $d=2, C=10$ and $d=2, C=3$. (c,d) CIFAR-100 with $d=64, C=100$ and $d=128, C=100$.

that vectors initialized with zero-mean Gaussian are uniformly distributed over the hypersphere (this phenomenon becomes more obvious in higher dimensions). To evaluate the intra-class feature variability, we consider a hyperspherical reverse-energy $E_r = \sum_{i \neq j \in A_c} \|\hat{x}_i - \hat{x}_j\|$ where $\hat{x}_i = \frac{x_i}{\|x_i\|}$ and $A_c$ denotes the sample index set of the $c$-th class. The smaller this reverse-energy gets, the less intra-class variability it implies. Figure 3(b,d) shows that the intra-class feature variability approaches to zero, as GNC(1) suggests. Details and more empirical results on GNC are in given Appendix A.

Now we discuss how to decouple the GNC hypothesis and how such a decoupling can enable us to design new objectives to train neural networks. GNC(1) and GNC(2) suggest to minimize intra-class feature variability and maximize inter-class feature separability, respectively. GNC(3) and GNC(4) are natural consequences if GNC(1) and GNC(2) hold. It has long been discovered in [42, 68, 82] that last-layer classifiers serve as proxies to represent the corresponding class of features, and they are also an approximation to the feature class-means. GNC(3) indicates the classifiers converge to hyperspherical uniformity, which, together with GNC(1), implies GNC(4).

Until now, it has been clear that GNC really boils down to two decoupled objectives: *maximize inter-class separability* and *minimize intra-class variability*, which again echos the goal of FDA. The problem reduces to how to effectively characterize these two objectives while being decoupled for flexibility (unlike CE or MSE). In the next section, we propose to address this problem by characterizing both objectives with a unified quantity - hyperspherical uniformity.

## 3 Hyperspherical Uniformity Gap

### 3.1 General Framework

As GNC(2) suggests, the inter-class separability is well captured by hyperspherical uniformity of feature class-means, so it is natural to directly use it as a learning target. On the other hand, GNC(1) does not suggest any easy-to-use quantity to characterize intra-class variability. We note that minimizing intra-class variability is actully equivalent to encouraging features of the same class to concentrate on a single point, which is the opposite of hyperspherical uniformity. Therefore, we can unify both intra-class variability and inter-class separability with a single characterization of hyperspherical uniformity. We propose to maximize the hyperspherical uniformity gap:

$$\max_{\{\hat{x}_j\}_{j=1}^n} \mathcal{L}_{\text{HUG}} := \alpha \cdot \underbrace{\mathcal{HU}(\{\hat{\mu}_c\}_{c=1}^C)}_{T_b: \text{ Inter-class Hyperspherical Uniformity}} - \beta \cdot \sum_{c=1}^C \underbrace{\mathcal{HU}(\{\hat{x}_i\}_{i \in A_c})}_{T_w: \text{ Intra-class Hyperspherical Uniformity}} \quad (2)$$

where $\alpha, \beta$ are hyperparameters, $\hat{\mu}_c = \frac{\mu_c}{\|\mu_c\|}$ is the feature class-mean projected on the unit hypersphere, $\mu_c = \sum_{c \in A_c} x_c$ is the feature class-mean, $x_i$ is the last-layer feature of the $i$-th sample and $A_c$ denotes the sample index set of the $c$-th class. $\mathcal{HU}(\{v_i\}_{i=1}^m)$ denotes some measure of hyperspherical uniformity for vectors $\{v_1, \cdots, v_m\}$. Eq. 2 is the general objective for HUG. Without loss of generality, we assume that the larger it gets, the stronger hyperspherical uniformity we have. We mostly focus on supervised learning with *parameteric class proxies*[2] where the CE loss is widely used as a *de facto* choice, although HUG can be used in much broader settings as discussed later. In the HUG framework, there is no longer a clear notion of classifiers (unlike the CE loss), but we still can utilize class proxies (*i.e.*, a generalized concept of classifiers) to facilitate the optimization.

We observe that Eq. 2 directly optimizes the feature class-means for inter-class separability, but they are intractable to compute during training (we need to compute them in every iteration). Therefore it is nontrivial to optimize the original HUG for training neural networks. A naive solution is to

---

[2]Parametric class proxies are a set of parameters used to represent a group of samples in the same class. Therefore, these proxies store the information about a class. Last-layer classifiers are a typical example.

approximate feature class-mean with a few mini-batches such that the gradients of $T_b$ can be still back-propagated to the last-layer features. However, it may take many mini-batches in order to obtain a sufficiently accurate class-mean, and the approximation gets much more difficult with large number of classes. To address this, we employ parametric class proxies to act as representatives of intra-class features and optimize them instead of feature class-means. We thus modify the HUG objective as

$$\max_{\{\hat{\boldsymbol{x}}_j\}_{j=1}^n, \{\hat{\boldsymbol{w}}_c\}_{c=1}^C} \mathcal{L}_{\text{P-HUG}} := \alpha \cdot \underbrace{\mathcal{H}\mathcal{U}\big(\{\hat{\boldsymbol{w}}_c\}_{c=1}^C\big)}_{\text{Inter-class Hyperspherical Uniformity}} - \beta \cdot \sum_{c=1}^C \underbrace{\mathcal{H}\mathcal{U}\big(\{\hat{\boldsymbol{x}}_i\}_{i \in A_c}, \hat{\boldsymbol{w}}_c\big)}_{\text{Intra-class Hyperspherical Uniformity}} \quad (3)$$

where $\hat{\boldsymbol{w}}_c \in \mathbb{S}^{d-1}$ is the parametric proxy for the $c$-th class. The intra-class hyperspherical uniformity term connects the class proxies with features by minimizing their joint hyperspherical uniformity, guiding features to move towards their corresponding class proxy. When training a neural network, the objective function in Eq. 3 will optimize network weights and proxies together. There are alternative ways to design the HUG loss from Eq. 2 for different learning scenarios, as discussed in Appendix C.

**Learnable proxies**. We can view the class proxy $\hat{\boldsymbol{\mu}}_i$ as learnable parameters and update them with stochastic gradients, similarly to the parameters of neural networks. In fact, learnable proxies play a role similar to the last-layer classifiers in the CE loss, improving the optimization by aggregating intra-class features. The major difference between learnable proxies and moving-averaged proxies is the way we update them. As GNC(3) implies, class proxies in HUG can also be used as classifiers.

**Static proxies**. Eq. 3 is decoupled into maximal inter-class separability and minimal intra-class variability. These two objects are independent and do not affect each other. We can thus optimize them independently. This suggests a even simpler way to assign class proxies – initializing class proxies with prespecified points that have attained hyperspherical uniformity, and fixing them in the training. There are two simple ways to obtain these class proxies: (1) minimizing their hyperspherical energy beforehand; (2) using zero-mean Gaussian to initialize the class proxies (Proposition 1). After initialization, class proxies will stay fixed and the features are optimized towards their class proxies.

**Partially learnable proxies**. After the class proxies are initialized using the static way above, we can increase its flexibility by learning an orthogonal matrix for the class proxies to find a suitable orientation for them. Specifically, we can learn this orthogonal matrix using methods in [47].

## 3.2 VARIATIONAL CHARACTERIZATION OF HYPERSPHERICAL UNIFORMITY

While there exist many ways to measure hyperspherical uniformity, we seek variational characterization due to simplicity. As examples, we consider minimum hyperspherical energy [45] that is inspired by Thomson problem [66, 70] and minimizes the potential energy, maximum hyperspherical separation [48] that is inspired by Tammes problem [69] and maximizes the smallest pairwise distance, and maximum gram determinant [48] that is defined by the volume of the formed parallelotope.

**Minimum hyperspherical energy**. MHE seeks to find an equilibrium state with minimum potential energy that distributes $n$ electrons on a unit hypersphere as evenly as possible. Hyperspherical uniformity is characterized by minimizing the hyperspherical energy for $n$ vectors $\boldsymbol{V}_n = \{\boldsymbol{v}_1, \cdots, \boldsymbol{v}_n \in \mathbb{R}^d\}$:

$$\min_{\{\hat{\boldsymbol{v}}_1, \cdots, \hat{\boldsymbol{v}}_n \in \mathbb{S}^{d-1}\}} \left\{ E_s(\hat{\boldsymbol{V}}_n) := \sum_{i=1}^n \sum_{j=1, j \neq i}^n K_s(\hat{\boldsymbol{v}}_i, \hat{\boldsymbol{v}}_j) \right\}, \quad K_s(\hat{\boldsymbol{v}}_i, \hat{\boldsymbol{v}}_j) = \left\{ \begin{array}{ll} \|\hat{\boldsymbol{v}}_i - \hat{\boldsymbol{v}}_j\|^{-s}, & s > 0 \\ -\|\hat{\boldsymbol{v}}_i - \hat{\boldsymbol{v}}_j\|^{-s}, & s < 0 \end{array} \right. , \quad (4)$$

where $\hat{\boldsymbol{v}}_i := \frac{\boldsymbol{v}_i}{\|\boldsymbol{v}_i\|}$ is the $i$-th vector projected onto the unit hypersphere. With $\mathcal{H}\mathcal{U}(\hat{\boldsymbol{V}}) = -E_s(\hat{\boldsymbol{V}})$, we apply MHE to HUG and formulate the new objective as follows ($s_b = 2, s_w = -1$):

$$\min_{\{\hat{\boldsymbol{x}}_j\}_{j=1}^n, \{\hat{\boldsymbol{w}}_c\}_{c=1}^C} \mathcal{L}_{\text{MHE-HUG}} := \alpha \cdot E_{s_b}\big(\{\hat{\boldsymbol{w}}_c\}_{c=1}^C\big) - \beta \cdot \sum_{c=1}^C E_{s_w}\big(\{\hat{\boldsymbol{x}}_i\}_{i \in A_c}, \hat{\boldsymbol{w}}_c\big) \quad (5)$$

which can already be used as to train neural networks. The intra-class variability term in Eq. 5 can be relaxed to a upper bound such that we can instead minimize a simple upper bound of $\mathcal{L}_{\text{MHE-HUG}}$:

$$\mathcal{L}'_{\text{MHE-HUG}} := \alpha \cdot \sum_{c \neq c'} \|\hat{\boldsymbol{w}}_c - \hat{\boldsymbol{w}}_{c'}\|^{-2} + \beta' \cdot \sum_c \sum_{i \in A_c} \|\hat{\boldsymbol{x}}_i - \hat{\boldsymbol{w}}_c\| \geq \mathcal{L}_{\text{MHE-HUG}} \quad (6)$$

which is much more efficient to compute in practice and thus can serve as a relaxed HUG objective. Moreover, $\mathcal{L}_{\text{MHE-HUG}}$ and $\mathcal{L}'_{\text{MHE-HUG}}$ share the same minimizer. Detailed derivation is in Appendix H.

**Maximum hyperspherical separation**. MHS uses a maximum geodesic separation principle by maximizing the *separation distance* $\vartheta(\hat{\boldsymbol{V}}_n)$ (*i.e.*, the smallest pairwise distance in $\boldsymbol{V}_n = \{\boldsymbol{v}_1, \cdots, \boldsymbol{v}_n \in$

$\mathbb{R}^d\})$: $\max_{\hat{\boldsymbol{V}}}\{\vartheta(\hat{\boldsymbol{V}}_n) := \min_{i \neq j}\|\hat{\boldsymbol{v}}_i - \hat{\boldsymbol{v}}_j\|\}$. Because $\vartheta(\hat{\boldsymbol{V}}_n)$ is another variational definition, we cannot naively set $\mathcal{HU}(\cdot) = \vartheta(\cdot)$. We define $\vartheta^{-1}(\hat{\boldsymbol{V}}_n) := \max_{i \neq j}\|\hat{\boldsymbol{v}}_i - \hat{\boldsymbol{v}}_j\|$ and HUG becomes

$$\max_{\{\hat{\boldsymbol{x}}_j\}_{j=1}^n, \{\hat{\boldsymbol{w}}_c\}_{c=1}^C} \mathcal{L}_{\text{MHS-HUG}} := \alpha \cdot \vartheta(\{\hat{\boldsymbol{w}}_c\}_{c=1}^C) - \beta \cdot \sum_{c=1}^C \vartheta^{-1}(\{\hat{\boldsymbol{x}}_i\}_{i \in A_c}, \hat{\boldsymbol{w}}_c), \tag{7}$$

which, by replacing intra-class variability with its surrogate, results in a more efficient form:

$$\mathcal{L}'_{\text{MHS-HUG}} := \alpha \cdot \min_{c \neq c'}\|\hat{\boldsymbol{w}}_c - \hat{\boldsymbol{w}}_{c'}\| - \beta \cdot \sum_c \max_{i \in A_c}\|\hat{\boldsymbol{x}}_i - \hat{\boldsymbol{w}}_c\| \tag{8}$$

which is a max-min optimization with a simple nearest neighbor problem inside. We note that $\mathcal{L}_{\text{MHS-HUG}}$ and $\mathcal{L}'_{\text{MHS-HUG}}$ share the same maximizer. Detailed derivation is given in Appendix H.

**Maximum gram determinant**. MGD characterizes the uniformity by computing a proxy to the volume of the parallelotope spanned by the vectors. MGD is defined with kernel gram determinant:

$$\max_{\{\hat{\boldsymbol{v}}_1, \cdots, \hat{\boldsymbol{v}}_n \in \mathbb{S}^{d-1}\}} \log \det \big(\boldsymbol{G} := \big(K(\hat{\boldsymbol{v}}_i, \hat{\boldsymbol{v}}_j)\big)_{i,j=1}^n\big), \quad K(\hat{\boldsymbol{v}}_i, \hat{\boldsymbol{v}}_j) = \exp\big(-\epsilon^2\|\hat{\boldsymbol{v}}_i - \hat{\boldsymbol{v}}_j\|^2\big) \tag{9}$$

where we use a Gaussian kernel with parameter $\epsilon$ and $\boldsymbol{G}(\hat{\boldsymbol{V}}_n)$ is the kernel gram matrix for $\hat{\boldsymbol{V}}_n = \{\hat{\boldsymbol{v}}_1, \cdots, \hat{\boldsymbol{v}}_n\}$. With $\mathcal{HU}(\hat{\boldsymbol{V}}_n) = \det \boldsymbol{G}(\hat{\boldsymbol{V}}_n)$, minimizing intra-class uniformity cannot be achieved by minimizing $\det \boldsymbol{G}(\hat{\boldsymbol{V}}_n)$, since $\det \boldsymbol{G}(\hat{\boldsymbol{V}}_n) = 0$ only leads to linear dependence. Then we have

$$\max_{\{\hat{\boldsymbol{x}}_j\}_{j=1}^n, \{\hat{\boldsymbol{w}}_c\}_{c=1}^C} \mathcal{L}_{\text{MGD-HUG}} := \alpha \cdot \log \det \big(\boldsymbol{G}(\{\hat{\boldsymbol{w}}_c\}_{c=1}^C)\big) + \beta' \cdot \sum_c \sum_{i \in A_c} \|\hat{\boldsymbol{x}}_i - \hat{\boldsymbol{w}}_c\| \tag{10}$$

where we directly use the surrogate loss from Eq. 6 as the intra-class variability term. With MGD, HUG has interesting geometric interpretation – it encourages the volume spanned by class proxies to be as large as possible and the volume spanned by intra-class features to be as small as possible.

### 3.3 Theoretical Insights and Discussions

There are many interesting theoretical questions concerning HUG, and this framework is highly related to a few topics in mathematics, such as tight frame theory [74], potential theory [39], sphere packing and covering [3, 18, 25]. The depth and breath of these topics are beyond imagination. In this section, we focus on discussing some highly related yet intuitive theoretical properties of HUG.

**Theorem 4 (Order of Minimum Hyperspherical Energy)** *If $d-1 > s > 0$ or $0 > s > -2$ and $d \in \mathbb{N}$, we have that $\lim_{n \to \infty}\{n^{-2} \cdot \min_{\hat{\boldsymbol{V}}_n} E_s(\hat{\boldsymbol{V}}_n)\} = c(s, d)$ where $c(s, d)$ is a constant involving $s, d$.*

The result above shows that the leading term of the minimum energy grows of order $\mathcal{O}(n^2)$ as $n \to \infty$. Theorem 4 generally holds with a wide range of $s$ for the Riesz kernel in hyperspherical energy. Moreover, the following result shows that MHS is in fact a limiting case of MHE as $s \to \infty$.

**Proposition 2 (MHS is a Limiting Case of MHE)** *Let $n \in \mathbb{N}, n \geq 2$ be fixed and $(\mathbb{S}^{d-1}, L_2)$ be a compact metric space. We have that $\lim_{s \to \infty}(\min_{\hat{\boldsymbol{V}}_n \subset \mathbb{S}^{d-1}} E_s(\hat{\boldsymbol{V}}_n))^{\frac{1}{s}} = (\max_{\hat{\boldsymbol{V}}_n \subset \mathbb{S}^{d-1}} \vartheta(\hat{\boldsymbol{V}}_n))^{-1}$.*

**Proposition 3** *The HUG objectives in both Eq. 5 and Eq. 6 converge to simplex ETF when $2 \leq C \leq d + 1$, converge to cross-polytope when $C = 2d$ and asymptotically converge to GNC as $C \to \infty$.*

Proposition 3 shows that HUG not only decouples GNC but also provably converges to GNC. Since GNC indicates that the CE loss eventually approaches to the maximizer of HUG, we now look into how the CE loss implicitly maximizes the HUG objective in a coupled way.

**Proposition 4** *The CE loss is $\mathcal{L}_{CE} = \sum_{i=1}^n \log(1 + \sum_{j \neq y_i}^C \exp(\langle \boldsymbol{w}_j, \boldsymbol{x}_i \rangle - \langle \boldsymbol{w}_{y_i}, \boldsymbol{x}_i \rangle))$ where $n$ is the number of samples, $\boldsymbol{x}_i$ is the $i$-th sample with label $y_i$ and $\boldsymbol{w}_j$ is the last-layer linear classifier for the $j$-th class. Bias is omitted for simplicity. $\mathcal{L}_{CE}$ is bounded by ($\rho = C - 1$)*

$$\underbrace{\sum_{i=1}^n \sum_{j \neq y_i}^C \langle \boldsymbol{w}_j, \boldsymbol{x}_i \rangle - \rho \underbrace{\sum_{i=1}^n \langle \boldsymbol{w}_{y_i}, \boldsymbol{x}_i \rangle}_{Q_2: \text{ Inter-class Variability}}}_{Q_1: \text{ Coupled IS and IV}} \leq \mathcal{L}_{CE} \leq \log\big(1 + \underbrace{\sum_{i=1}^n \sum_{j \neq y_i}^C \exp(\langle \boldsymbol{w}_j, \boldsymbol{x}_i \rangle)}_{Q_3: \text{ Coupled IS and IV}} + \rho \underbrace{\sum_{i=1}^n \exp(-\langle \boldsymbol{w}_{y_i}, \boldsymbol{x}_i \rangle)}_{Q_4: \text{ Inter-class Variability}}\big).$$

We show in Proposition 4 that CE inherently optimizes two independent criterion: intra-class variability (IV) and inter-class separability (IS). With normalized classifiers and features, we can see that $Q_1$ and $Q_3$ have similar minimum where $\boldsymbol{x}_i = \boldsymbol{w}_{y_i}$ and $\boldsymbol{w}_i, \forall i$ attain hyperspherical uniformity.

We show that CE is lower bounded by the gap of inter-class and intra-class hyperspherical uniformity:

$$\mathcal{L}_{\text{CE}} \geq \underbrace{\sum_{i=1}^{n} \log \sum_{c=1}^{C} \exp(\rho_2 \sum_{j=1}^{n} l_{jc} \langle \boldsymbol{x}_i, \boldsymbol{x}_j \rangle) - \rho_3 \sum_{i=1}^{n} \left\| \frac{1}{n} \sum_{i=1}^{n} l_{ic} \boldsymbol{x}_i \right\|^2}_{\text{Inter-class Hyperspherical Uniformity}} - \underbrace{\rho_1 \sum_{i=1}^{n} \sum_{j \in A_{y_i}} \langle \boldsymbol{x}_i, \boldsymbol{x}_j \rangle}_{\text{Intra-class Hyperspherical Uniformity}} \quad (11)$$

where $\rho_1, \rho_2, \rho_3$ are constants and $l_{ic}$ is the softmax confidence of $\boldsymbol{x}_i$ for the $c$-th class (Appendix L). This result [4] implies that minimizing CE effectively minimizes HUG. [50] proves that the minimizer of the normalized CE loss converges to hyperspherical uniformity. We rewrite their results below:

**Theorem 5 (CE Asymptotically Converges to HUG's Maximizer)** *Considering unconstrained features of $C$ classes (each class has the same number of samples), with features and classifiers normalized on some hypersphere, we have that, for the minimizer of the CE loss, classifiers converge weakly to the uniform measure on $\mathbb{S}^{d-1}$ as $C \to \infty$ and features collapse to their corresponding classifiers. The minimizer of CE also asymptotically converges to the maximizer of HUG.*

Theorem 5 shows that the minimizer of the CE loss with unconstrained features [53] asymptotically converges to the maximizer of HUG (*i.e.*, GNC). Till now, we show that HUG shares the same optimum with CE (with hyperspherical normalization), while being more flexible for decoupling inter-class feature separability and intra-class feature variability. Therefore, we argue that HUG can be an excellent alternative for the widely used CE loss in classification problems.

**HUG maximizes mutual information**. We can view HUG as a way to maximize mutual information $\mathcal{I}(\boldsymbol{X}; \boldsymbol{Y}) = \mathcal{H}(\boldsymbol{X}) - \mathcal{H}(\boldsymbol{X}|\boldsymbol{Y})$, where $\boldsymbol{X}$ denotes the feature space and $\boldsymbol{Y}$ is the label space. Maximizing $\mathcal{H}(\boldsymbol{X})$ implies that the feature should be uniform over the space. Minimizing $\mathcal{H}(\boldsymbol{X}|\boldsymbol{Y})$ means that the feature from the same class should be concentrated. This is nicely connected to HUG.

**The role of feature and class proxy norm**. Both NC and GNC do not take the norm of feature and class proxy into consideration. HUG also assume both feature and class proxy norm are projected onto some hypersphere. Although dropping these norms usually improves generalizability [8, 9, 14, 43, 76], training neural networks with standard CE loss still yields different class proxy norms and feature norms. We hypothesize that this is due to the underlying difference among training data distribution of different classes. One empirical evidence to support this is that average feature norm of different classes is consistent across training under different random seeds (*e.g.*, average feature norm for digit 1 on MNIST stays the smallest in different run). [36, 46, 52] empirically show that feature norm corresponds to the quality of the sample, which can also viewed as a proxy to sample uncertainty. [56] theoretically shows that the norm of neuron weights (*e.g.*, classifier) matters for its Rademacher complexity. As a trivial solution to minimize the CE loss, increasing the classifier norm (if the feature is correctly classified) can easily decrease the CE loss to zero for this sample, which is mostly caused by the softmax function. Taking both feature and class proxy norm into account greatly complicates the analysis (*e.g.*, it results in weighted hyperspherical energy where the potentials between vectors are weighted) and seem to yield little benefit for now. We defer this issue to future investigation.

**HUG as a general framework for designing loss functions**. HUG can be viewed as an inherently decoupled way of designing new loss functions. As long as we design a measure of hyperspherical uniformity, then HUG enables us to effortlessly turn it into a loss function for neural networks.

## 4 EXPERIMENTS AND RESULTS

Our experiments aims to demonstrate the empirical effectiveness of HUG, so we focus on the fair comparison to the popular CE loss under the same setting. Experimental details are in Appendix N.

### 4.1 EXPLORATORY EXPERIMENTS AND ABLATION STUDY

**Different HUG variants**. We compare different HUG variants and the CE loss on CIFAR-10 and CIFAR-100 with ResNet-18 [29]. Specifically, we use Eq. 6, Eq. 6 and Eq. 10 for MHE-HUG, MHS-HUG and MGD-HUG, respectively. The results are given in Table 1. We can observe that all HUG variants outperform the CE loss. Among all, MHE-HUG achieves the best testing accuracy with considerable improvement over the CE loss. We note that all HUG variants are

| Method | CIFAR-10 | CIFAR-100 |
|---|---|---|
| CE Loss | 5.45 | 24.90 |
| MHE-HUG | **5.03** | **23.50** |
| MHS-HUG | 5.09 | 24.38 |
| MGD-HUG | 5.38 | 24.59 |

Table 1: Testing error (%) of HUG variants on CIFAR-10 and CIFAR-100.

used without the CE loss. The performance gain of HUG are actually quite significant, since the CE loss is currently a default choice for classification problems and serves as a very strong baseline.

**Different methods to update proxies**. We also evaluate how different proxy update methods will affect the classification performance. We use the same setting as Table 1. For all the proxy update methods, we apply them to MHE-HUG (Eq. 6) under the same setting. The results are given Table 2. We can observe that all the propose proxy update methods work reasonably well. More interestingly, static proxies work surprisingly well and outperform the CE loss even when all the class proxies are randomly

| Method | CIFAR-10 | CIFAR-100 |
|---|---|---|
| CE Loss | 5.45 | 24.90 |
| Fully learnable | **5.03** | **23.50** |
| Static (random) | 5.19 | 24.23 |
| Static (optimized) | 5.12 | 24.02 |
| Partially learnable | 5.08 | 23.89 |

Table 2: Testing error (%) of different proxy update methods on CIFAR-10 and CIFAR-100.

initialized and then fixed throughout the training. The reason the static proxies work for MHE-HUG is due to Proposition 1. This result is significant since we no longer have to train class proxies in HUG (unlike CE). When trained with large number of classes, it is GPU-memory costly for learning class proxies, which is also known as one of the bottlenecks for face recognition [1]. HUG could be a promising solution to this problem.

**Loss landscape and convergence**. We perturb neuron weights (refer to [40]) to visualize the loss landscape of HUG and CE in Figure 4. We use MHE in HUG here. The results show that HUG yields much flatter

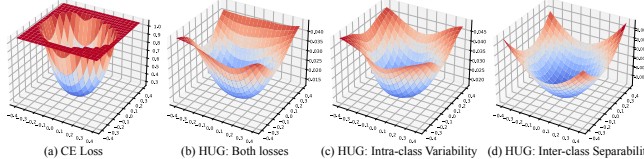

(a) CE Loss  (b) HUG: Both losses  (c) HUG: Intra-class Variability  (d) HUG: Inter-class Separability

Figure 4: Loss landscape visualization. (b,c,d) show $\mathcal{L}'_{\text{MHE-HUG}}$, $T_b$ and $T_w$, respectively.

local minima than the CE loss in general, implying that HUG has potentially stronger generalization [34, 57]. We show more visualizations and convergence dynamics in Appendix O.

**Learning with different architectures**. We evaluate HUG with different network architectures such as VGG-16 [65], ResNet-18 [29] and DenseNet-121 [31]. Results in Table 3 (Left number: CIFAR-10, right number: CIFAR-100) show

| Method | ResNet-18 | VGG-16 | DenseNet-121 |
|---|---|---|---|
| CE Loss | 5.45 / 24.90 | 5.28 / 22.99 | 5.04 / 21.47 |
| HUG | **5.03 / 23.50** | **5.19 / 22.77** | **4.85 / 21.30** |

Table 3: Testing error (%) with different architectures.

that HUG is agnostic to different network architectures and outperforms the CE loss in every case. Although HUG works well on its own, any other methods that improve CE can also work with HUG.

## 4.2 GENERALIZATION AND ROBUSTNESS UNDER DIFFERENT LEARNING SCENARIOS

**Long-tailed recognition**. We consider the task of long-tailed recognition, where the data from different classes are imbalanced. The settings generally follow [6], and the dataset gets more imbalanced if the imbalance ratio (IR) gets smaller. The potential

| | CIFAR-100 | | | | CIFAR-10 | | | |
|---|---|---|---|---|---|---|---|---|
| IR | 0.2 | 0.1 | 0.02 | 0.01 | 0.2 | 0.1 | 0.02 | 0.01 |
| CE | 66.74 | 62.31 | 48.79 | 43.82 | 90.29 | 87.85 | 79.17 | 74.11 |
| HUG | **67.83** | **63.33** | **50.48** | **45.63** | **90.41** | **88.20** | **79.88** | **75.14** |

Table 4: Testing accuracy (%) of long-tailed recognition.

of HUG in imbalanced classification is evident, as the inter-class separability in the HUG is explicitly modeled and can be easily controlled. Experimental results in Table 4 show that HUG can consistently outperform the CE loss in the challenging long-tailed setting under different imbalanced ratio.

**Continual learning**. We demonstrate the potential of HUG in the class-continual learning setting, where the training data is not sampled *i.i.d.* but comes in class by class. Since training data is highly biased, hyperspherical uniformity among class proxies is crucial. Due to

| | CIFAR-100 | | | CIFAR-10 | | |
|---|---|---|---|---|---|---|
| Memory size | 200 | 500 | 2000 | 200 | 500 | 2000 |
| ER + CE | 22.14 | 31.02 | 43.54 | 49.07 | 61.58 | 76.89 |
| ER + HUG | **23.52** | **31.92** | **43.92** | **53.74** | **62.67** | **77.21** |

Table 5: Final testing accuracy (%) of continual learning.

the decoupled nature of HUG, we can easily increase the importance of inter-class separability, unlike CE. We use a simple continual learning method – ER [62] where the CE loss with memory is used. We replace it with HUG. Table 5 shows HUG consistently improves ER under different memory size.

**Adversarial robustness**. We further test HUG's adversarial robustness. In our experiments, we consider the classical white-box PGD attack [51] on ResNet-18. The PGD attack iteration is set as 100 and the attack strength level is set as 2/255, 4/255, 8/255 in

| Method | Clean | $l_\infty$=2/255 | $l_\infty$=4/255 | $l_\infty$=8/255 |
|---|---|---|---|---|
| CE Loss | 5.45 / 24.90 | 7.94 / 2.12 | 0.61 / 0 | 0 / 0 |
| HUG | **5.03 / 23.50** | **15.24 / 5.26** | **3.45 / 1.24** | **1.76 / 0.44** |

Table 6: Testing accuracy (%) under adversarial attacks.

$l_\infty$ norm. All networks are naturally training with either HUG or CE loss. Results in Table 6 demonstrate that HUG yields consistently stronger adversarial robustness than the CE loss.

**NLP tasks**. As an exploration, we evaluate HUG on some simple NLP classification tasks. Our experiments follow the same settings as [32] and finetune the BERT model [15] in these tasks. Table 7 shows that HUG yields better generalizability than CE, demonstrating its potential for NLP.

| Task | MRPC | SST-2 | WNLI |
|---|---|---|---|
| CE Loss | 84.8 | 91.6 | 33.8 |
| HUG | **85.8** | **91.8** | **34.0** |

Table 7: NLP testing accuracy (%)

## 5 RELATED WORK AND CONCLUDING REMARKS

We start by generalizing and decoupling the NC phenomenon, obtaining two basic principles for loss functions. Based on these principles, we identify a quantity hyperspherical uniformity gap, which not only decouples NC but also provides a general framework for designing loss functions. We demonstrate a few simple HUG variants that outperform the CE loss in terms of generalization and adversarial robustness. There is a large body of excellent work in NC that is related to HUG, such as [26, 33, 71, 89]. [88] extends the study of NC to more practical loss functions (*e.g.*, focal loss and losses with label smoothing). Different from existing work in hyperspherical uniformity [41, 45, 48] and generic diversity (decorrelation) [2, 7, 11, 54, 77, 83], HUG works as a new learning target (used without CE) rather than acting as a regularizer for the CE loss (used together with CE). Following the spirit of [32], we demonstrate the effectiveness and potential of HUG as a valid substitute for CE.

**Relevant theoretical results**. [87] has discussed NC under the case of $d < C - 1$, and shown that the global solution in this case yields the best rand-$d$ approximation of the simplex ETF. Along with [87], GNC gives a more profound characterization of the convergence of class-means. We show a special case of $d = 2, C = 4$. It is easy to see that hyperspherical uniformity in this case forms four vectors with adjacency ones being perpendicular. This is also the case captured by the best rank-2 approximation (*i.e.*, a 2-dimensional hyperplane with simplex ETF projected onto it). Figure 5 gives a geometric interpretation for the connection between [87] and GNC. [3] provides an in-depth introduction and comprehensive theoretical analysis for the energy minimization problem, which significantly benefits this work.

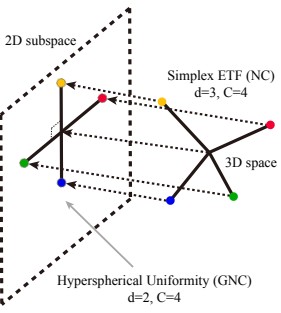

Figure 5: Geometric connection between GNC and [87].

**Connection to contrastive learning**. The goal of contrastive learning [8, 10, 24, 30, 72, 78, 85] is to learn discriminative features through instance-wise discrimination and contrast. Despite the lack of class labels, [78] discovers that contrastive learning performs sample-wise alignment and sample-wise uniformity, sharing a similar high-level spirit to intra-class variability and inter-class separability. [35] adapts contrastive learning to the supervised settings where labeled samples are available, which also shares conceptual similarity to our framework and settings.

**Related work on (deep) metric learning**. Metric learning also adopts similar idea where similar samples are pulled together and dissimilar ones are pushed away. HUG has intrinsic connections to a number of loss functions in metric learning [4, 17, 21, 24, 58, 59, 61, 67, 68, 79, 79–81, 84].

## 6 BROADER IMPACT AND FUTURE WORK

Our work reveals the underlying principle – hyperspherical uniformity gap, for classification loss function, especially in the context of deep learning. We provide a simple yet effective framework for designing decoupled classification loss functions. Rather than previous objective functions that are coupled and treated as a black-box, our loss function has clear physical interpretation and is fully decoupled for different functionalities. These characteristics may help neural networks to identify intrinsic structures hidden in data and true causes for classifying images. HUG may have broader applications in interpretable machine learning and fairness / bias problems.

Our work is by no means perfect, and there are many aspects that require future investigation. For example, the implicit data mining in CE [49] is missing in the current HUG design, current HUG losses are more sensitive to hyperparameters than CE (the flexibility of decoupling also comes at a price), current HUG losses could be more unstable to train (more difficult to converge) than CE, and it requires more large-scale experiments to fully validate the superiority of current HUG losses. We hope that our work can serve as a good starting point to rethink classification losses in deep learning.

## ACKNOWLEDGEMENT

The authors would like to sincerely thank the anonymous reviewers for all the detailed and valuable suggestions that have significantly improved the paper. This work is supported by the German Federal Ministry of Education and Research (BMBF): Tübingen AI Center, FKZ: 01IS18039A, 01IS18039B; and by the Machine Learning Cluster of Excellence, EXC number 2064/1 – Project number 390727645. AW acknowledges support from a Turing AI Fellowship under EPSRC grant EP/V025279/1, and the Leverhulme Trust via CFI.

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

# Appendix

## Table of Contents

## A EMPIRICAL RESULTS ON GENERALIZED NEURAL COLLAPSE

### A.1 DETAILED METRIC DEFINITION

We consider four metrics: average classifier energy (ACE), average class-mean energy (ACME), average feature reverse-energy (AFRE) and average feature-mean reverse-energy (AFMRE) in the paper. Their definitions are given below:

$$E_{\text{ACE}} = \frac{1}{C(C-1)} \sum_{i \neq j} \|\hat{\boldsymbol{w}}_i - \hat{\boldsymbol{w}}_j\|^{-2} \tag{12}$$

$$E_{\text{ACME}} = \frac{1}{C(C-1)} \sum_{i \neq j} \|\hat{\boldsymbol{\mu}}_i - \hat{\boldsymbol{\mu}}_j\|^{-2} \tag{13}$$

$$E_{\text{AFRE}} = \frac{1}{C} \sum_{c=1}^{C} \frac{1}{|A_c| \cdot (|A_c| - 1)} \sum_{i \neq j \in A_c} \|\hat{\boldsymbol{x}}_i - \hat{\boldsymbol{x}}_j\| \tag{14}$$

$$E_{\text{AFMRE}} = \frac{1}{C} \sum_{c=1}^{C} \frac{1}{|A_c|} \sum_{i \in A_c} \|\hat{\boldsymbol{x}}_i - \hat{\boldsymbol{\mu}}_c\| \tag{15}$$

where $|A_c|$ denotes the cardinality of the set $A_c$, $\hat{\boldsymbol{\mu}}_c$ is the normalized feature mean of the $c$-th class and $\hat{\boldsymbol{w}}_c$ denotes the normalized class proxy of the $c$-th class.

### A.2 EMPIRICAL RESULTS OF GNC ON IMAGENET

We find that the GNC hypothesis remains valid and informative even under the scenario of large number of classes (we use the 1000-class ImageNet-2012 dataset [13] here). Experimental results with ResNet-18 [29] (feature dimension as 512) are given in Figure 6. Experimental results with ResNet-50 [29] (feature dimension as 2048) are given in Figure 7.

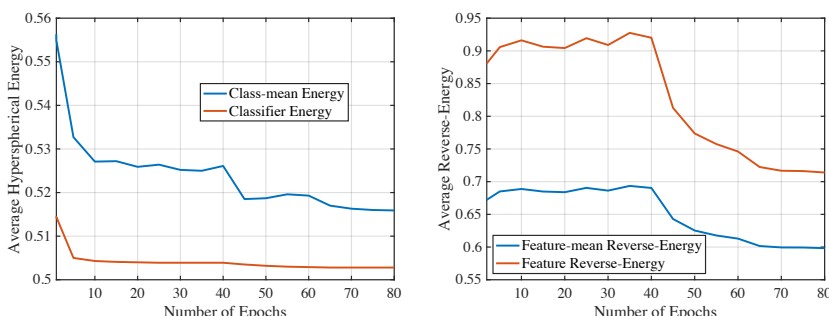

Figure 6: Training dynamics of hyperspherical energy (which captures inter-class separability) and hyperspherical reverse-energy (which captures intra-class variability). ImageNet-2012 [13] with ResNet-18 [29] ($d = 512$, $C = 1000$).

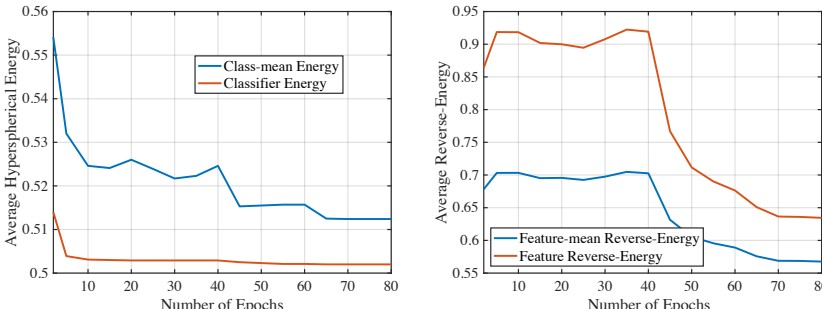

Figure 7: Training dynamics of hyperspherical energy (which captures inter-class separability) and hyperspherical reverse-energy (which captures intra-class variability). ImageNet-2012 [13] with ResNet-50 [29] ($d = 2048$, $C = 1000$).

# B  2D MNIST FEATURE VISUALIZATION

We also visualize the 2D MNIST feature in Figure 8, Figure 9 and Figure 10, which is done by directly setting the output feature dimension as 2. Different color denotes different class and black arrow denotes the class proxy. We compare the difference between the CE loss and the HUG-MHE loss (with either independently optimized proxies or fully learnable proxies). Specifically, for the HUG-MHE loss with independently optimized proxies, we use the following form:

$$\max_{\{\hat{\boldsymbol{x}}_j\}_{j=1}^n, \{\hat{\boldsymbol{w}}_c\}_{c=1}^C} \mathcal{L}_{\text{P-HUG}} := \alpha \cdot \underbrace{\mathcal{HU}\left(\{\hat{\boldsymbol{w}}_c\}_{c=1}^C\right)}_{\text{Inter-class Hyperspherical Uniformity}} - \beta \cdot \sum_{c=1}^C \underbrace{\mathcal{HU}\left(\{\hat{\boldsymbol{x}}_i\}_{i \in A_c}, \text{SG}(\hat{\boldsymbol{w}}_c)\right)}_{\text{Intra-class Hyperspherical Uniformity}} \quad (16)$$

where we stop the gradient for the class proxies in the intra-class hyperspherical uniformity term. Form the results, we observe that the our HUG losses generally learns better representations than the CE loss, and moreover, HUG learns more aligned class proxy and class feature-mean than CE.

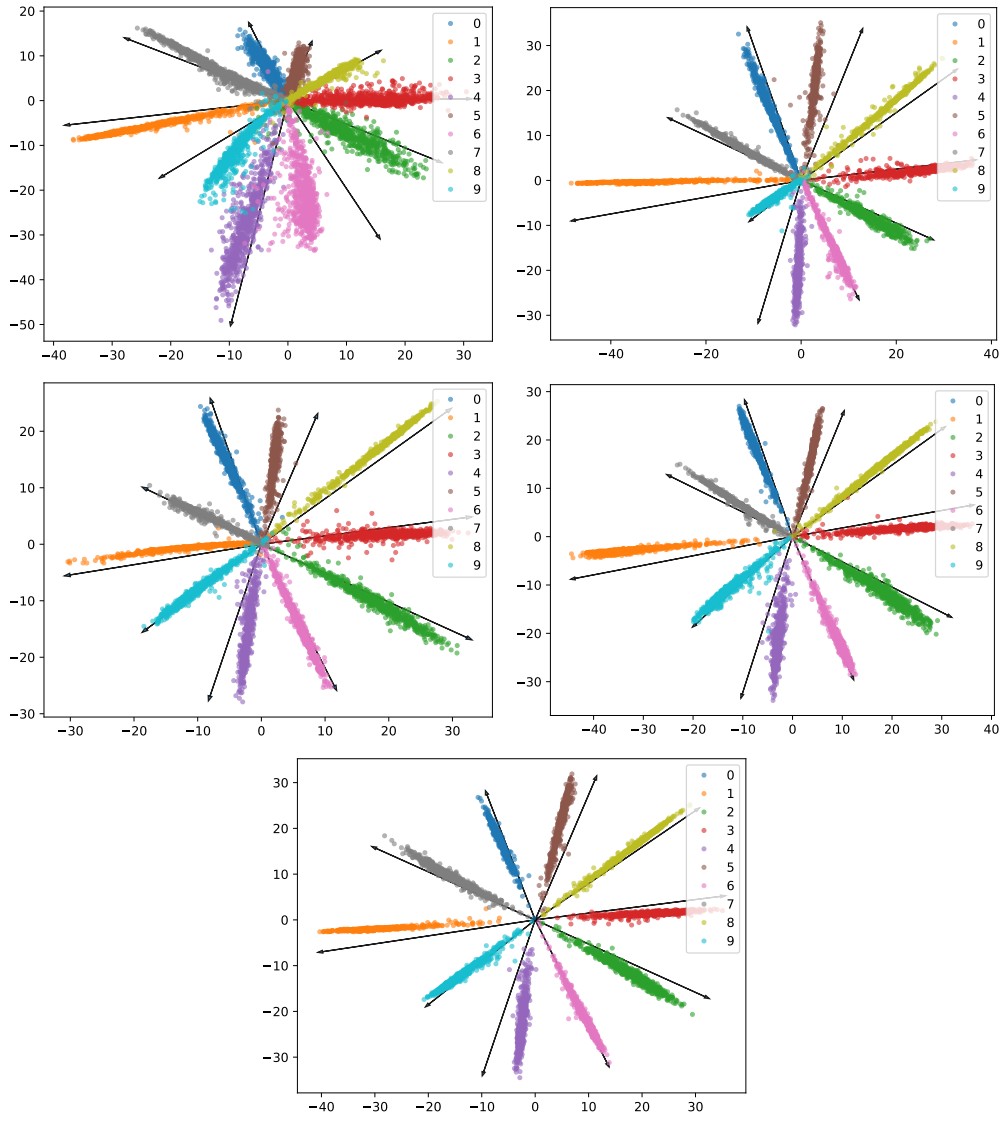

Figure 8: 2D MNIST feature visualization for the CE loss at 1,5,10,15,20 epochs (top left - top right - middle left - middle right -bottom).

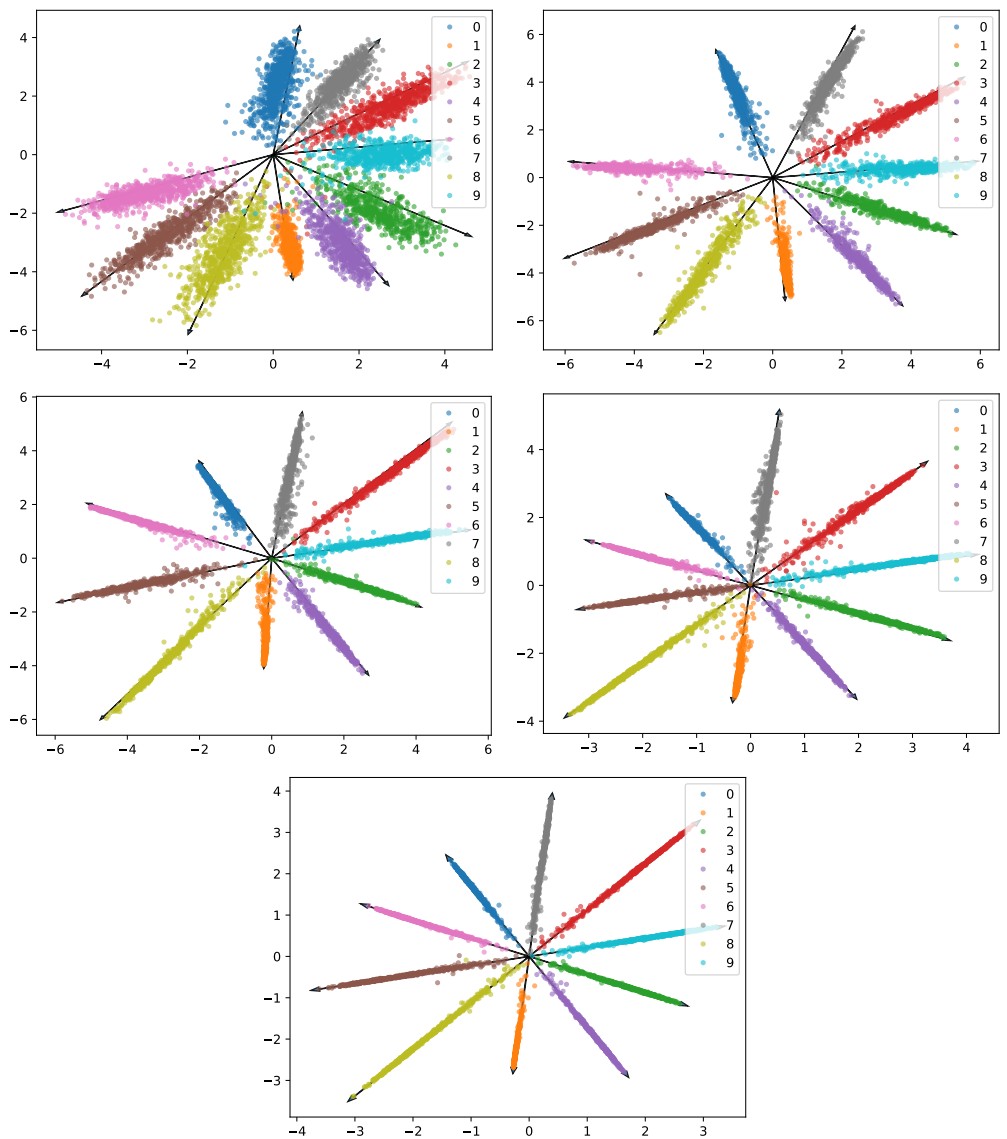

Figure 9: 2D MNIST feature visualization for the HUG loss (randomly initialized and then optimized proxies) at 1,5,10,15,20 epochs (top left - top right - middle left - middle right -bottom).

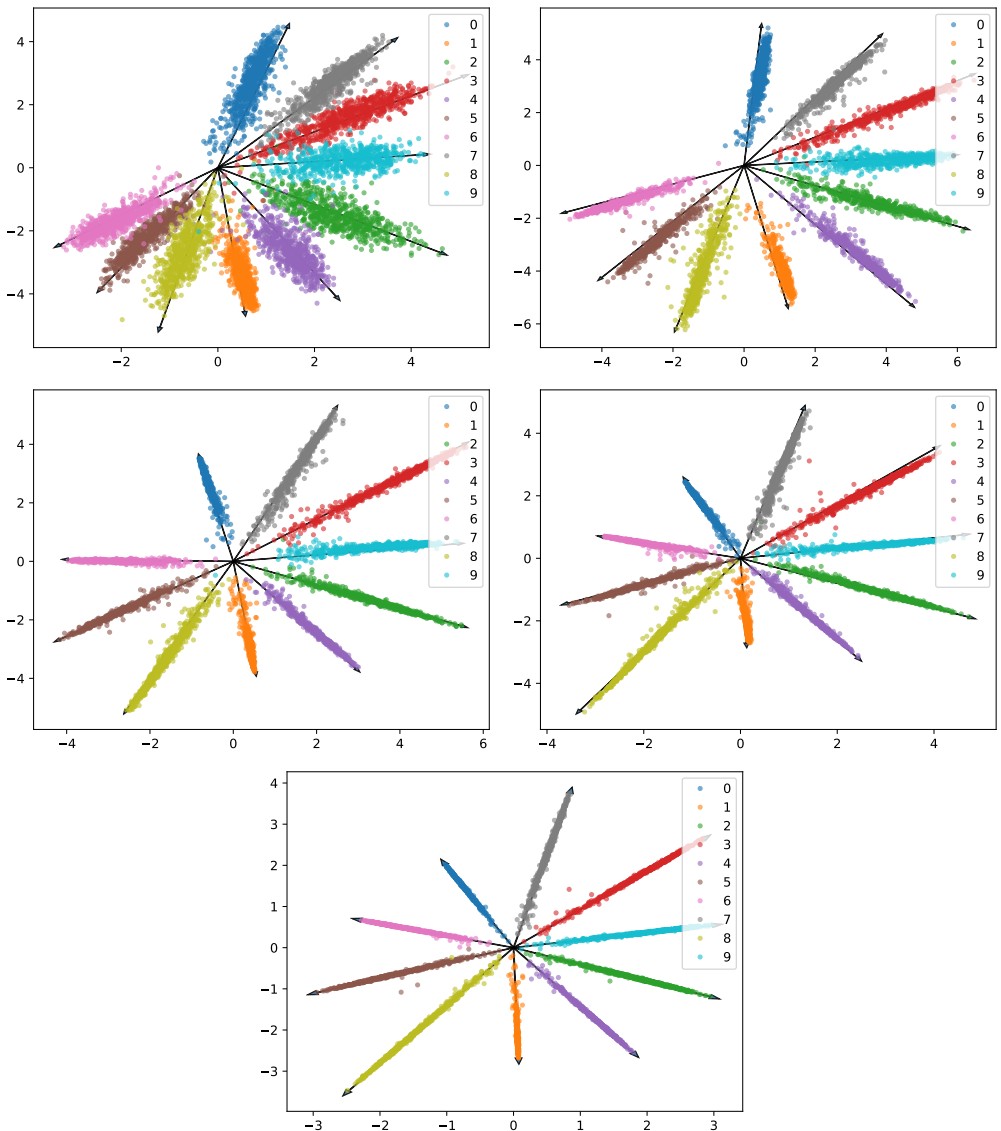

Figure 10: 2D MNIST feature visualization for the HUG loss (fully learnable proxies) at 1,5,10,15,20 epochs (top left - top right - middle left - middle right -bottom).

# C OTHER VARIANTS IN THE HUG FRAMEWORK

There are plenty of interesting and useful instantiations for the loss function under the HUG framework. In this section, we discuss a few highly relevant and natural ones.

## C.1 PROXY-FREE HUG

We have the following general HUG objective function:

$$\max_{\{\hat{\boldsymbol{x}}_j\}_{j=1}^n} \mathcal{L}_{\text{HUG}} := \alpha \cdot \underbrace{\mathcal{HU}\big(\{\hat{\boldsymbol{\mu}}_c\}_{c=1}^C\big)}_{T_b: \text{ Inter-class Hyperspherical Uniformity}} - \beta \cdot \sum_{c=1}^{C} \underbrace{\mathcal{HU}\big(\{\hat{\boldsymbol{x}}_i\}_{i\in A_c}\big)}_{T_w: \text{ Intra-class Hyperspherical Uniformity}} \tag{17}$$

where we can have many possible instantiations. Other than the proxy-based form proposed in the main paper, we can also have a proxy-free version:

$$\max_{\{\hat{\boldsymbol{x}}_j\}_{j=1}^n} \mathcal{L}_{\text{PF-HUG}} := \alpha \cdot \underbrace{\mathcal{HU}\big(\{\hat{\boldsymbol{x}}_{i\in A_c}\}_{c=1}^C\big)}_{\text{Inter-class Hyperspherical Uniformity}} - \beta \cdot \sum_{c=1}^{C} \underbrace{\mathcal{HU}\big(\{\hat{\boldsymbol{x}}_i\}_{i\in A_c}\big)}_{\text{Intra-class Hyperspherical Uniformity}} \tag{18}$$

where $\{\hat{\boldsymbol{x}}_{i\in A_c}\}_{c=1}^C$ denotes a set of vectors that consist of one random sample per class. This is essentially to replace the class proxy with a random sample from this class. The proxy-free HUG loss can be used in the scenario where extremely large amount of classes exist and storing class proxies can be very expensive, or in the scenario of self-supervised contrastive learning where each instance and its augmentations are viewed as one class. A MHE-based instantiation of Eq. 18 is given by

$$\min_{\{\hat{\boldsymbol{x}}_j\}_{j=1}^n} \mathcal{L}_{\text{MHE-PF-HUG}} := \alpha \cdot E_{s_b}\big(\{\hat{\boldsymbol{x}}_{i\in A_c}\}_{c=1}^C\big) - \beta \cdot \sum_{c=1}^{C} E_{s_w}\big(\{\hat{\boldsymbol{x}}_i\}_{i\in A_c}\big) \tag{19}$$

which can be similarly relaxed to

$$\mathcal{L}'_{\text{MHE-PF-HUG}} = \alpha \cdot \sum_{c\neq c'} \|\hat{\boldsymbol{x}}_{i\in A_c} - \hat{\boldsymbol{x}}_{j\in A_{c'}}\|^{-2} + \beta' \cdot \sum_c \sum_{i\in A_c, j\in A_c, i\neq j} \|\hat{\boldsymbol{x}}_i - \hat{\boldsymbol{x}}_j\| \tag{20}$$

where $\hat{\boldsymbol{x}}_{i\in A_c}$ denotes a randomly selected sample from the $c$-th class. The first term in Eq. 20 can also be viewed as a scalable stochastic approximation to the first term in the following loss function:

$$\mathcal{L}''_{\text{MHE-PF-HUG}} = \alpha \cdot \sum_{i\in A_c, j\in A_{c'}, c\neq c'} \|\hat{\boldsymbol{x}}_i - \hat{\boldsymbol{x}}_j\|^{-2} + \beta' \cdot \sum_c \sum_{i\in A_c, j\in A_c, i\neq j} \|\hat{\boldsymbol{x}}_i - \hat{\boldsymbol{x}}_j\| \tag{21}$$

which is typically optimized by stochastic gradients (samples come as a mini batch) in practice.

## C.2 COUPLED HUG

One advantage of HUG is that it decouples intra-class variability and inter-class separability. However, coupling may also bring some benefits (*e.g.*, robustness on hyperparameters, stability in training). To this end, we also propose a coupled loss function using the HUG framework:

$$\max_{\{\hat{\boldsymbol{x}}_j\}_{j=1}^n} \mathcal{L}_{\text{PF-HUG}} := \alpha \cdot \underbrace{\sum_{i=1}^{n} \mathcal{HU}\big(\{\hat{\boldsymbol{w}}_c\}_{c=1, c\neq y_i}^C, \hat{\boldsymbol{x}}_i\big)}_{\text{Coupled Intra-class and Inter-class Hyperspherical Uniformity}} - \beta \cdot \sum_{c=1}^{C} \underbrace{\mathcal{HU}\big(\{\hat{\boldsymbol{x}}_i\}_{i\in A_c}\big)}_{\text{Intra-class Hyperspherical Uniformity}} \tag{22}$$

which can be turned into a MHE-based instantiation:

$$\mathcal{L}''_{\text{MHE-C-HUG}} = \alpha \cdot \sum_{i=1}^{n} \sum_{c\neq y_i} \|\hat{\boldsymbol{x}}_i - \hat{\boldsymbol{w}}_c\|^{-2} + \beta' \cdot \sum_c \sum_{i\in A_c, j\in A_c, i\neq j} \|\hat{\boldsymbol{x}}_i - \hat{\boldsymbol{x}}_j\| \tag{23}$$

where the first term itself couples intra-class and inter-class hyperspherical uniformity. Although the coupled HUG drops the flexibility that the original HUG framework brings, it may introduce extra advantages (*e.g.*, training stability).

## C.3 HUG without Hyperspherical Normalization

While the CE loss does not necessarily require hyperspherical normalization for the proxies and features (but hyperspherical normalization does improve CE's generalizability [44, 75]), we also consider the HUG framework without hyperspherical normalization here. We note that this issue remains an open challenge and we only aim to provide some simple yet natural designs.

The obvious problem to remove hyperspherical normalization is that HUG has a trivial way to decrease its loss – simply increasing the magnitude of features and proxies. A naive way to address this is to introduce magnitude penalty terms for the features and proxies. This results in

$$
\max_{\{\boldsymbol{x}_j\}_{j=1}^n, \{\boldsymbol{w}_c\}_{c=1}^C} \mathcal{L}_{\text{UN-P-HUG}} := \alpha \cdot \underbrace{\mathcal{HU}\big(\{\boldsymbol{w}_c\}_{c=1}^C\big)}_{\text{Inter-class Hyperspherical Uniformity}} - \beta \cdot \sum_{c=1}^C \underbrace{\mathcal{HU}\big(\{\boldsymbol{x}_i\}_{i \in A_c}, \boldsymbol{w}_c\big)}_{\text{Intra-class Hyperspherical Uniformity}}
$$

$$
- \lambda_1 \cdot \underbrace{\sum_{c=1}^C \big\| \boldsymbol{w}_c - s \big\|^2}_{\text{Soft Magnitude Constraint on Proxies}} - \lambda_2 \cdot \underbrace{\sum_{i=1}^n \big\| \boldsymbol{x}_i - s \big\|^2}_{\text{Soft Magnitude Constraint on Features}}
$$

where $s$ denotes the magnitude hyperparameter.

## D  PROOF OF THEOREM 1

We first let $\hat{V}_C = \{\hat{v}_1, \cdots, \hat{v}_C\}$ be an arbitrary vector configuration in $\mathbb{S}^{d-1}$. Then we will have that

$$
\begin{aligned}
\Lambda(\hat{V}_C) &:= \sum_{i=1}^{C} \sum_{j=1}^{C} \|\hat{v}_i - \hat{v}_j\|^2 \\
&= \sum_{i=1}^{C} \sum_{j=1}^{C} (2 - 2\hat{v}_i \cdot \hat{v}_j) \\
&= 2C^2 - 2 \left\| \sum_{i=1}^{C} \hat{v}_i \right\|^2 \\
&\leq 2C^2
\end{aligned}
\tag{24}
$$

which holds if and only if $\sum_{i=1}^{C} \hat{v}_i = 0$. The vertices of a regular $(n-1)$-simplex at the origin well satisfy this condition. With the properties of the potential function $f$, we have that

$$
\begin{aligned}
E_f(\hat{v}_C) &:= \sum_{i=1}^{C} \sum_{j:j\neq i} f\left( \|\hat{v}_i - \hat{v}_j\|^2 \right) \\
&\geq C(C-1) f\left( \frac{\Lambda(\hat{v}_C)}{C(C-1)} \right) \\
&\geq C(C-1) f\left( \frac{2C}{C-1} \right)
\end{aligned}
\tag{25}
$$

which holds true if all pairwise distance $\|\hat{v}_i - \hat{v}_j\|$ are equal for $i \neq j$ and the center of mass is at the origin (*i.e.*, $\sum_{i=1}^{C} \hat{v}_i = 0$). Therefore, for the vector configuration $\hat{V}_C^*$ which contains the vertices of a regular $(C-1)$-simplex inscribed in $\mathbb{S}^d$ and centered at the origin, we have that for $2 \leq C \leq d+1$

$$
\begin{aligned}
E_f(\hat{V}_n^*) &= C(C-1) f\left( \frac{2C}{C-1} \right) \\
&\leq E_f(\hat{V}_C).
\end{aligned}
\tag{26}
$$

If $f$ is strictly convex and strictly decreasing, then $E_f(\hat{V}_C) \geq C(C-1) f(\frac{2C}{C-1})$ holds only when $\hat{V}_C^*$ is a regular $(C-1)$-simplex inscribed in $\mathbb{S}^{d-1}$ and centered at the origin. ∎

# E    PROOF OF THEOREM 2

This result comes as a natural conclusion from [12] where they prove that any sharp code is a minimal hyperspherical $f$-energy $N$-point configuration for any interaction potential $f$ that is absolutely monotone on $[-1, 1]$ including all Riesz $s$-potentials $f(t) = 2(t - 2t)^{-s/2}$ for $s > 0$.

Before we move on, we need to introduce the definition of sharp code:

**Definition 1** *Let $\hat{V}_N = \{\hat{v}_1, \cdots, \hat{v}_N\}$ be a $N$-point configuration on $\mathbb{S}^{d'}$.*

- *If for every $(d' + 1)$-variate polynomial $P$ of degree at most $m$,*

$$\int_{\mathbb{S}^{d'}} P d\sigma_{d'} = \frac{1}{N} \sum_{i=1}^{N} P(\hat{v}_i)$$

  *then $\hat{V}_N$ is called a spherical $m$-design.*

- *If $\hat{V}_N$ is a configuration of $N$ distinct points such that the set of inner products between distinct points in $\hat{V}_N$ has cardinality $k$, then $\hat{V}_N$ is called a spherical $k$-distance set.*

- *The configuration $\hat{V}_N$ is a sharp code if it is both a $k$-distance set and a spherical $(2k - 1)$-design.*

The Cohn-Kumar Universal Optimality theorem [12] states that any sharp code is universally optimal. By universal optimality, we mean that

**Definition 2** *An $N$-point configuration $\hat{V}_N$ on $\mathbb{S}^{d'}$ is called universally optimal if*

$$E_f(\hat{V}_N) := \sum_{\hat{v}_1, \hat{v}_2 \in \hat{V}_N, \hat{v}_1 \neq \hat{v}_2} f(\hat{v}_1^\top \hat{v}_2) = \min_{\hat{V}_N \subset \mathbb{S}^{d'}} E_f(\hat{V}_N)$$

*holds for any absolutely monotone function $f : [-1, 1) \to \mathbb{R}$.*

Then formally, Cohn-Kuma Universal Optimality Theorem states:

**Theorem 6** *If $\hat{V}_N$ is a sharp code on $\mathbb{S}^{d'}$, then $\hat{V}_N$ is universally optimal.*

Because the vertices of the cross-polytope are a sharp code, then this vertex set ($2d' + 2$ points in total) is universally optimal, which implies that

$$E_f(\hat{W}_N) = \min_{\hat{W}_N \subset \mathbb{S}^{d'}} E_f(\hat{W}_N) \tag{27}$$

where $\hat{W}_N$ denote the vertex set of the cross-polytope. Then we let $s = 2$ for the $f$-energy and $d' = d - 1$, and we prove our theorem. ∎

## F  PROOF OF THEOREM 3

This theorem is in fact a well-known result (see [3, 27, 38, 63]). This general result is stated as

**Theorem 7** *If $A \subset \mathbb{R}^p$ is compact with $\dim A > 0$ and $0 < s < \dim A$, then*

$$\lim_{N \to \infty} \frac{\varepsilon_s(A, N)}{N^2} = W_s(A),$$

*where $\varepsilon_s(A, n) := \min_{\hat{\boldsymbol{W}}_n \subset A} E_s(\hat{\boldsymbol{W}}_n)$ and $W_s(A)$ is Wiener constant. Moreover, the equilibrium measure $\mu_{s,A}$ on $A$ is unique for the Riesz $s$-kernel when $0 < s < \dim A$. Finally, any sequence $\{\hat{\boldsymbol{v}}_1^N, \cdots, \hat{\boldsymbol{v}}_N^N\}_{N=2}^{\infty}$ of asympototically $s$-energy minimizing $N$-point configuration on $A$ satisfies*

$$v(\{\hat{\boldsymbol{v}}_1^N, \cdots, \hat{\boldsymbol{v}}_N^N\}) \to_{weak} \mu_{s,A}, \ N \to \infty$$

From the theorem above, with $s = 2$, $d - 1 > s$, $N = C$ and $A = \mathbb{S}^{d-1}$, we have that $W_s(\mathbb{S}^{d-1})$ is a constant term, and most importantly, we have that these point sequences $\{\hat{\boldsymbol{\mu}}_1^C, \cdots, \hat{\boldsymbol{\mu}}_C^C\}$ asymptotically minimizes the hyperspherical energy on $\mathbb{S}^{d-1}$.

Moreover, the same theorem also gives that the leading term of the minimum hyperspherical energy is of order $\mathcal{O}(n^2)$ as $n \to \infty$. ∎

# G  PROOF OF PROPOSITION 1

We show that zero-mean equal-variance Gaussian distributed vectors (after normalized to norm 1) are uniformly distributed over the unit hypersphere with Theorem 8.

**Lemma 1** *Let $\boldsymbol{x}$ be a $n$-dimensional random vector with distribution $\mathcal{N}(0,1)$ and $\boldsymbol{U} \in \mathbb{R}^{n \times n}$ be an orthogonal matrix ($\boldsymbol{U}\boldsymbol{U}^\top = \boldsymbol{U}^\top\boldsymbol{U} = \boldsymbol{I}$). Then $\boldsymbol{Y} = \boldsymbol{U}\boldsymbol{x}$ also has the distribution of $\mathcal{N}(0,1)$.*

**Proof G.1** *For any measurable set $A \subset \mathbb{R}^n$, we have that*

$$
\begin{aligned}
P(Y \in A) &= P(X \in U^\top A) \\
&= \int_{U^\top A} \frac{1}{(\sqrt{2\pi})^n} e^{-\frac{1}{2}\langle x, x\rangle} \\
&= \int_A \frac{1}{(\sqrt{2\pi})^n} e^{-\frac{1}{2}\langle Ux, Ux\rangle} \\
&= \int_A \frac{1}{(\sqrt{2\pi})^n} e^{-\frac{1}{2}\langle x, x\rangle}
\end{aligned}
\tag{28}
$$

*because of orthogonality of $U$. Therefore the lemma holds.* ∎

**Theorem 8** *The normalized vector of Gaussian variables is uniformly distributed on the sphere. Formally, let $x_1, x_2, \cdots, x_n \sim \mathcal{N}(0,1)$ and be independent. Then the vector*

$$
\boldsymbol{x} = \left[\frac{x_1}{z}, \frac{x_2}{z}, \cdots, \frac{x_n}{z}\right]
\tag{29}
$$

*follows the uniform distribution on $\mathbb{S}^{n-1}$, where $z = \sqrt{x_1^2 + x_2^2 + \cdots + x_n^2}$ is a normalization factor.*

**Proof G.2** *A random variable has distribution $\mathcal{N}(0,1)$ if it has the density function*

$$
f(x) = \frac{1}{\sqrt{2\pi}} e^{-\frac{1}{2}x^2}.
\tag{30}
$$

*A $n$-dimensional random vector $\boldsymbol{x}$ has distribution $\mathcal{N}(0,1)$ if the components are independent and have distribution $\mathcal{N}(0,1)$ each. Then the density of $\boldsymbol{x}$ is given by*

$$
f(x) = \frac{1}{(\sqrt{2\pi})^n} e^{-\frac{1}{2}\langle x, x\rangle}.
\tag{31}
$$

*Then we use Lemma 1 about the orthogonal-invariance of the normal distribution.*

*Because any rotation is just a multiplication with some orthogonal matrix, we know that normally distributed random vectors are invariant to rotation. As a result, generating $\boldsymbol{x} \in \mathbb{R}^n$ with distribution $\mathbb{N}(0,1)$ and then projecting it onto the hypersphere $\mathbb{S}^{n-1}$ produces random vectors $U = \frac{\boldsymbol{x}}{\|\boldsymbol{x}\|}$ that are uniformly distributed on the hypersphere. Therefore the theorem holds.* ∎

The above results indicate that as long as class proxies are initialize with zero-mean Gaussian, they are uniformly distributed over the hypersphere in a probabilistic sense. ∎

## H  DERIVATION OF HUG SURROGATE FOR MHE AND MHS

The derivation of $\mathcal{L}'_{\text{MHE-HUG}}$ is as follows:

$$
\begin{aligned}
\mathcal{L}_{\text{MHE-HUG}} &:= \alpha \cdot E_{s_b}\big(\{\hat{\boldsymbol{w}}_c\}_{c=1}^C\big) - \beta \cdot \sum_{c=1}^C E_{s_w}\big(\{\hat{\boldsymbol{x}}_i\}_{i\in A_c}, \hat{\boldsymbol{w}}_c\big) \\
&= \alpha \cdot \sum_{c\neq c'} \|\hat{\boldsymbol{w}}_c - \hat{\boldsymbol{w}}_{c'}\|^{-2} + \beta \cdot \sum_c \big( \sum_{i,j\in A_c, i\neq j} \|\hat{\boldsymbol{x}}_i - \hat{\boldsymbol{x}}_j\| + 2\cdot \sum_{i\in A_c} \|\hat{\boldsymbol{x}}_i - \hat{\boldsymbol{w}}_c\|\big) \\
&= \alpha \cdot \sum_{c\neq c'} \|\hat{\boldsymbol{w}}_c - \hat{\boldsymbol{w}}_{c'}\|^{-2} + \beta \cdot \sum_c \big( \sum_{i,j\in A_c, i\neq j} \|\hat{\boldsymbol{x}}_i - \hat{\boldsymbol{w}}_c + \hat{\boldsymbol{w}}_c - \hat{\boldsymbol{x}}_j\| \\
&\qquad\qquad\qquad\qquad\qquad\qquad\qquad\qquad\qquad\qquad + 2\cdot \sum_{i\in A_c} \|\hat{\boldsymbol{x}}_i - \hat{\boldsymbol{w}}_c\|\big) \qquad (32) \\
&\leq \alpha \cdot \sum_{c\neq c'} \|\hat{\boldsymbol{w}}_c - \hat{\boldsymbol{w}}_{c'}\|^{-2} + \beta \cdot \sum_c \big( \sum_{i,j\in A_c, i\neq j} (\|\hat{\boldsymbol{x}}_i - \hat{\boldsymbol{w}}_c\| + \|\hat{\boldsymbol{w}}_c - \hat{\boldsymbol{x}}_j\|) \\
&\qquad\qquad\qquad\qquad\qquad\qquad\qquad\qquad\qquad\qquad + 2\cdot \sum_{i\in A_c} \|\hat{\boldsymbol{x}}_i - \hat{\boldsymbol{w}}_c\|\big) \\
&= \alpha \cdot \sum_{c\neq c'} \|\hat{\boldsymbol{w}}_c - \hat{\boldsymbol{w}}_{c'}\|^{-2} + \beta' \cdot \sum_c \sum_{i\in A_c} \|\hat{\boldsymbol{x}}_i - \hat{\boldsymbol{w}}_c\| =: \mathcal{L}'_{\text{MHE-HUG}}
\end{aligned}
$$

The derivation of $\mathcal{L}'_{\text{MHS-HUG}}$ is as follows:

$$
\begin{aligned}
\mathcal{L}_{\text{MHS-HUG}} &:= \alpha \cdot \vartheta\big(\{\hat{\boldsymbol{w}}_c\}_{c=1}^C\big) - \beta \cdot \sum_{c=1}^C \vartheta\big(\{\hat{\boldsymbol{x}}_i\}_{i\in A_c}, \hat{\boldsymbol{w}}_c\big) \\
&= \alpha \cdot \min_{c\neq c'} \|\hat{\boldsymbol{w}}_c - \hat{\boldsymbol{w}}_{c'}\| - \beta \cdot \sum_c \max_{\boldsymbol{u},\boldsymbol{v}\in\{\{\hat{\boldsymbol{x}}_i\}_{i\in A_c}, \hat{\boldsymbol{w}}_c\}, \boldsymbol{u}\neq\boldsymbol{v}} \|\boldsymbol{u} - \boldsymbol{v}\| \qquad (33) \\
&\leq \alpha \cdot \min_{c\neq c'} \|\hat{\boldsymbol{w}}_c - \hat{\boldsymbol{w}}_{c'}\| - \beta \cdot \sum_c \max_{i\in A_c} \|\hat{\boldsymbol{x}}_i - \hat{\boldsymbol{w}}_c\| =: \mathcal{L}'_{\text{MHS-HUG}}.
\end{aligned}
$$

Most importantly, $\sum_c \max_{\boldsymbol{u},\boldsymbol{v}\in\{\{\hat{\boldsymbol{x}}_i\}_{i\in A_c}, \hat{\boldsymbol{w}}_c\}, \boldsymbol{u}\neq\boldsymbol{v}} \|\boldsymbol{u} - \boldsymbol{v}\|$ in $\mathcal{L}_{\text{MHS-HUG}}$ and $\mathcal{L}'_{\text{MHS-HUG}}$ share the same minimizer (minimum is 0, which happens when intra-class feature collapse to its class proxy). Therefore, $\mathcal{L}'_{\text{MHS-HUG}}$ and $\mathcal{L}_{\text{MHS-HUG}}$ share the same maximizer, and $\mathcal{L}'_{\text{MHS-HUG}}$ can be viewed as a surrogate loss for $\mathcal{L}_{\text{MHS-HUG}}$.

# I    PROOF OF PROPOSITION 2

For notational convenience, we first define $\varepsilon_s(\mathbb{S}^{d-1}, n) := \min_{\hat{\boldsymbol{V}}_n \subset \mathbb{S}^{d-1}} E_s(\hat{\boldsymbol{V}}_n)$ and $\delta_n^\rho(\mathbb{S}^{d-1}) := \max_{\hat{\boldsymbol{V}}_n \subset \mathbb{S}^{d-1}} \vartheta(\hat{\boldsymbol{V}}_n)$. We then define that $\hat{\boldsymbol{V}}_n^s$ is a $s$-energy minimizing $n$-point configuration on $\mathbb{S}^{d-1}$ if $0 < s < \infty$ (*i.e.*, MHE configuration) and $\hat{\boldsymbol{V}}_n^\infty$ denotes a best-packing configuration on $\mathbb{S}^{d-1}$ if $s = \infty$ (*i.e.*, MHS configuration). Since we are considering $s > 0$, we only need to discuss the case of $K_s(\hat{\boldsymbol{v}}_i, \hat{\boldsymbol{v}}_j) = \rho(\hat{\boldsymbol{v}}_i, \hat{\boldsymbol{v}}_j)^{-s}$. Then we will have the following equation:

$$\varepsilon_s(\mathbb{S}^{d-1}, n)^{\frac{1}{s}} = E_s(\hat{\boldsymbol{V}}_n^s)^{\frac{1}{s}} \geq \frac{1}{\delta_n^\rho(\hat{\boldsymbol{V}}_n^s)} \geq \frac{1}{\delta_n^\rho(\mathbb{S}^{d-1})}. \tag{34}$$

Moreover, we have that

$$
\begin{aligned}
\varepsilon_s(\mathbb{S}^{d-1}, n)^{\frac{1}{s}} &\leq E_s(\hat{\boldsymbol{V}}_n^\infty)^{\frac{1}{s}} \\
&= \frac{1}{\delta^\rho(\hat{\boldsymbol{V}}_n^\infty)} \left( \sum_{1 \leq i \neq j \leq N} \left( \frac{\delta^\rho(\hat{\boldsymbol{V}}_n^\infty)}{\rho(\hat{\boldsymbol{v}}_i^\infty, \hat{\boldsymbol{v}}_j^\infty)} \right)^s \right)^{\frac{1}{s}} \\
&\leq \frac{1}{\delta^\rho(\hat{\boldsymbol{V}}_n^\infty)} \left( n(n-1) \right)^{\frac{1}{s}}
\end{aligned}
\tag{35}
$$

Therefore, we will end up with

$$\lim_{s \to \infty} \sup \varepsilon_s(\mathbb{S}^{d-1}, n)^{\frac{1}{s}} \leq \frac{1}{\delta^\rho(\hat{\boldsymbol{V}}_n^\infty)} = \frac{1}{\delta_n^\rho(\mathbb{S}^{d-1})}. \tag{36}$$

Then we take both Eq. 34 and Eq. 36 into consideration and have that

$$\lim_{s \to \infty} \varepsilon_s(\mathbb{S}^{d-1}, n)^{\frac{1}{s}} = \frac{1}{\delta_n^\rho(\mathbb{S}^{d-1})} \tag{37}$$

which concludes the proof. ∎

## J  PROOF OF PROPOSITION 3

We write down the formulation of the HUG objectives (with MHE):

$$
\min_{\{\hat{\boldsymbol{x}}_i\}_{i=1}^n, \{\hat{\boldsymbol{w}}_c\}_{c=1}^C} \mathcal{L}_{\text{MHE-HUG}} := \alpha \cdot E_{s_b}\big(\{\hat{\boldsymbol{w}}_c\}_{c=1}^C\big) - \beta \cdot \sum_{c=1}^C E_{s_w}\big(\{\hat{\boldsymbol{x}}_i\}_{i \in A_c}, \hat{\boldsymbol{w}}_c\big)
$$

$$
= \alpha \cdot \sum_{c \neq c'} \|\hat{\boldsymbol{w}}_c - \hat{\boldsymbol{w}}_{c'}\|^{-2} + \beta \cdot \sum_c \Big( \sum_{i,j \in A_c, i \neq j} \|\hat{\boldsymbol{x}}_i - \hat{\boldsymbol{x}}_j\| \tag{38}
$$

$$
+ 2 \cdot \sum_{i \in A_c} \|\hat{\boldsymbol{x}}_i - \hat{\boldsymbol{w}}_c\| \Big)
$$

$$
\min_{\{\hat{\boldsymbol{x}}_i\}_{i=1}^n, \{\hat{\boldsymbol{w}}_c\}_{c=1}^C} \mathcal{L}'_{\text{MHE-HUG}} = \alpha \cdot \sum_{c \neq c'} \|\hat{\boldsymbol{w}}_c - \hat{\boldsymbol{w}}_{c'}\|^{-2} + \beta' \cdot \sum_c \sum_{i \in A_c} \|\hat{\boldsymbol{x}}_i - \hat{\boldsymbol{w}}_c\| \tag{39}
$$

For both objectives, we can see that the minimizer of the second term (*i.e.*, the intra-class variability term) is all intra-class feature collapse to their class proxy and therefore the second term achieves the global minimum $0$.

For the first term of both objectives, the global minimizer can be obtain directly from Theorem 1, Theorem 2 and Theorem 3. It is easy to see that the global minimizer of the inter-class separability term and the intra-class variability term does not contradict with each other and can be achieved simultaneously. ∎

## K    PROOF OF PROPOSITION 4

$$\sum_{i=1}^{n} \log(1 + \sum_{j=1 \neq y_i}^{C} \exp(\langle \boldsymbol{w}_j, \boldsymbol{x}_i \rangle - \langle \boldsymbol{w}_{y_i}, \boldsymbol{x}_i \rangle))$$

$$\geq \sum_{i=1}^{n} \sum_{j=1 \neq y_i}^{C} \log(1 + \exp(\langle \boldsymbol{w}_j, \boldsymbol{x}_i \rangle - \langle \boldsymbol{w}_{y_i}, \boldsymbol{x}_i \rangle))$$

$$\geq \sum_{i=1}^{n} \sum_{j=1 \neq y_i}^{C} (\langle \boldsymbol{w}_j, \boldsymbol{x}_i \rangle - \langle \boldsymbol{w}_{y_i}, \boldsymbol{x}_i \rangle) \tag{40}$$

$$= \underbrace{\sum_{i=1}^{n} \sum_{j \neq y_i}^{C} \langle \boldsymbol{w}_j, \boldsymbol{x}_i \rangle}_{Q_1: \text{ Coupling IS and IV}} - \underbrace{(C-1) \sum_{i=1}^{n} \langle \boldsymbol{w}_{y_i}, \boldsymbol{x}_i \rangle}_{Q_2: \text{ Inter-class Variability}}$$

$$\sum_{i=1}^{n} \log(1 + \sum_{j=1 \neq y_i}^{C} \exp(\langle \boldsymbol{w}_j, \boldsymbol{x}_i \rangle - \langle \boldsymbol{w}_{y_i}, \boldsymbol{x}_i \rangle))$$

$$\leq \log(1 + \sum_{i=1}^{n} \sum_{j=1 \neq y_i}^{C} \exp(\langle \boldsymbol{w}_j, \boldsymbol{x}_i \rangle - \langle \boldsymbol{w}_{y_i}, \boldsymbol{x}_i \rangle))$$

$$\leq \log(1 + \sum_{i=1}^{n} \sum_{j=1 \neq y_i}^{C} (\exp(\langle \boldsymbol{w}_j, \boldsymbol{x}_i \rangle) + \exp(-\langle \boldsymbol{w}_{y_i}, \boldsymbol{x}_i \rangle))) \tag{41}$$

$$= \log \Big(1 + \underbrace{\sum_{i=1}^{n} \sum_{j \neq y_i}^{C} \exp(\langle \boldsymbol{w}_j, \boldsymbol{x}_i \rangle)}_{Q_3: \text{ Coupling IS and IV}} + \underbrace{(C-1) \sum_{i=1}^{n} \exp(-\langle \boldsymbol{w}_{y_i}, \boldsymbol{x}_i \rangle)}_{Q_4: \text{ Inter-class Variability}} \Big)$$

∎

## L  DERIVATION OF CE'S LOWER BOUND

The derivation is actually very simple and this result is originally given by [4]. We find that it naturally matches the intuition behind HUG. For our paper to be self-contained, we briefly give the simple derivation below. For the details, please refer to Proposition 1 in [4].

We start by rewriting the CE loss as

$$\mathcal{L}_{\text{CE}} = \underbrace{-\sum_{i=1}^{n} \langle \boldsymbol{w}_{y_i}, \boldsymbol{x}_i \rangle + \frac{\lambda n}{2} \sum_{c=1}^{C} \langle \boldsymbol{w}_c, \boldsymbol{w}_c \rangle}_{Q_1(\boldsymbol{w})} + \underbrace{\sum_{i=1}^{n} \log \sum_{c=1}^{C} \exp(\langle \boldsymbol{w}_c, \boldsymbol{x}_i \rangle) - \frac{\lambda n}{2} \sum_{c=1}^{C} \langle \boldsymbol{w}_c, \boldsymbol{w}_c \rangle}_{Q_2(\boldsymbol{w})} \quad (42)$$

where $\lambda$ can be chosen such that both $Q_1(\boldsymbol{w})$ and $Q_2(\boldsymbol{w})$ become convex functions with respect to $\boldsymbol{w}$. Taking advantage of the convexity, we can separately set the gradient of $Q_1(\boldsymbol{w})$ and $Q_2(\boldsymbol{w})$ with respect to $\boldsymbol{w}$ as 0 and compute their minima. Specifically, we end up with

$$Q_1(\boldsymbol{w}) \geq Q_1(\boldsymbol{w}_{Q_1}^*) = -\frac{1}{2\lambda n} \sum_{i=1}^{n} \sum_{j \in A_{y_i}} \langle \boldsymbol{x}_i, \boldsymbol{x}_j \rangle, \quad (43)$$

$$Q_2(\boldsymbol{w}) \geq Q_2(\boldsymbol{w}_{Q_2}^*) = \sum_{i=1}^{n} \log \sum_{c=1}^{C} \exp\left(\frac{1}{\lambda n} \sum_{j=1}^{n} l_{jc} \langle \boldsymbol{x}_i, \boldsymbol{x}_j \rangle\right) - \frac{n}{2\lambda} \sum_{c=1}^{C} \left\| \frac{1}{n} \sum_{i=1}^{n} l_{ic} \boldsymbol{x}_i \right\|^2, \quad (44)$$

where $l_{ic} = \frac{\exp(\langle \boldsymbol{w}_c, \boldsymbol{w}_i \rangle)}{\sum_j \exp(\langle \boldsymbol{w}_j, \boldsymbol{x}_i \rangle)}$ denotes the softmax confidence. Combining the two lower bounds above, we can have that

$$\mathcal{L}_{\text{CE}} \geq Q_1(\boldsymbol{w}_{Q_1}^*) + Q_2(\boldsymbol{w}_{Q_2}^*)$$

$$= \sum_{i=1}^{n} \log \sum_{c=1}^{C} \exp\left(\frac{1}{\lambda n} \sum_{j=1}^{n} l_{jc} \langle \boldsymbol{x}_i, \boldsymbol{x}_j \rangle\right) - \frac{n}{2\lambda} \sum_{c=1}^{C} \left\| \frac{1}{n} \sum_{i=1}^{n} l_{ic} \boldsymbol{x}_i \right\|^2 - \frac{1}{2\lambda n} \sum_{i=1}^{n} \sum_{j \in A_{y_i}} \langle \boldsymbol{x}_i, \boldsymbol{x}_j \rangle$$

$$(45)$$

where the first two terms encourage larger inter-class hyperspherical uniformity, and the last term promotes smaller intra-class hyperspherical uniformity.

# M PROOF OF THEOREM 5

This theorem follows naturally from the main result in [50]. [50] has proved that the minimizer of a simplified form of the cross-entropy loss is the simplex ETF when $2 \leq C \leq d+1$ and the minimizer also asymptotically converges to uniform measure on the hypersphere. More formally, we have

**Theorem 9 ([50])** *Consider the following variational problem*

$$\min_{\boldsymbol{u}} \mathcal{L}_{\alpha}(\boldsymbol{u}) := \sum_{i=1}^{n} \log \left( \frac{\sum_{j=1}^{n} \exp(\langle \boldsymbol{u}_j, \boldsymbol{u}_i \rangle)}{\exp(\langle \boldsymbol{u}_i, \boldsymbol{u}_i \rangle)} \right) \tag{46}$$
$$\text{s.t. } \boldsymbol{u}_i \in \mathbb{R}^d, \|\boldsymbol{u}_i\| = 1, \forall i$$

*Let $\mu_n$ be the probability measure on $\mathbb{S}^d$ generated by a minimizer*

$$\mu_n = \frac{1}{n} \sum_{i=1}^{n} \delta_{\boldsymbol{u}_i}, \tag{47}$$

*then for any $\alpha > 0$, $\mu_n$ converges weakly to the uniform measure on $\mathbb{S}^{d-1}$ as $n \to \infty$.*

From Theorem 3, we know that HUG with specific potential energy also converges to the uniform measure on $\mathbb{S}^{d-1}$. Combining the results above, we can conclude that HUG and CE share the same minimizer. ∎

# N   EXPERIMENTAL DETAILS

**General settings**. For MHE-HUG and MHS-HUG, $\alpha$ and $\beta$ are set as 0.15 and 0.015, respectively. For MGD-HUG, $\alpha$ and $\beta$ are set as 0.15 and 0.03, respectively. We train the model for 200 epochs with 512 batchsize for both the cross-entropy (CE) loss and HUG. We use the stochastic gradient descent with momentum 0.9 and weight decay $2 \times 10^{-4}$. The initial learning rate is set as 0.1 for both CIFAR-100 and CIFAR-10 and is divided by 10 at 60, 120, 180 epoch. For the general classification experiments, we use multiple architectures, including ResNet-18, VGG16 and DenseNet121. we use the simple data augmentation: 4 pixels are padded on each side, and image is randomly cropped.

**Long-tailed recognition.** We follow LDAM [6] to obtain imbalanced CIFAR-10 and CIFAR-100 datasets with different imbalanced ratio. Following LDAM, we use ResNet-32 as our base network. The other setting is the same as our general setting.

**Continual learning.** We follow DER [5] to construct our continual learning experiments. We split both the CIFAR-10 and CIFAR-100 training set into 5 tasks. Each task has 2 classes and 20 classes for CIFAR-10 and CIFAR-100, respectively. The training batchsize is set as 64, where there are 32 incoming samples and 32 replayed samples. Different size of memory buffer is also studied.

**Adversarial robustness.** For the experiments of adversarial robustness, we first obtain the model trained with CE and HUG. With the information of the attacked model, PGD [51] generates some adversarial examples to mislead the attacked model. The test accuracy in the experiments of adversarial robustness shows the accuracy of the perturbed samples.

**Visualizing loss landscape.** We perturb neuron weights to visualize the loss landscape, as proposed in [40]. For details, we perturb the model weight with 400 interpolation points in two random vectors around the current model weight minima. The visualization method is also the same as [47].

# O   ADDITIONAL EXPERIMENTAL RESULTS

**Training convergence.** We observe the training convergence of HUG on CIFAR-10 and CIFAR-100. Both the evaluation accuracy and the training loss, including the overall losses, the intra-class loss and the inter-class loss, are shown in Figure 11. For both the CIFAR-10 and CIFAR-100, the inter-class uniformity loss remains relatively small, which is consistent with the empirical finding in [41, 47]. Moreover, we find that the intra-class uniformity loss (*i.e.*, intra-class variability) dominates the overall loss on CIFAR-100 dataset and it is relatively difficult to optimize when the class number becomes large.

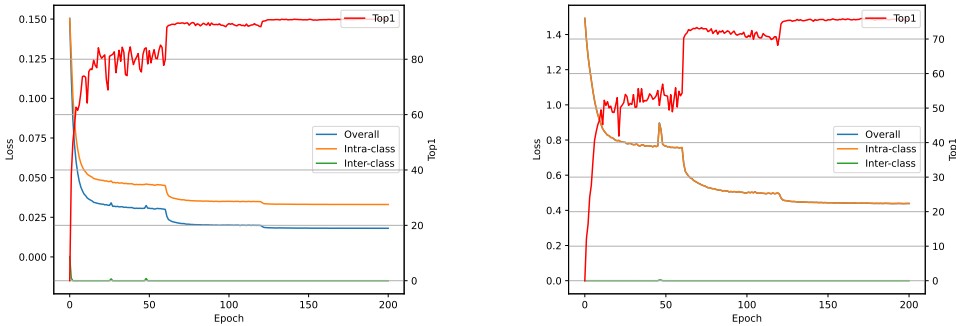

Figure 11: HUG's training loss and testing accuracy (%) on CIFAR-10 (left) and CIFAR-100 (right).

**2D loss contour.** We also utilize the method in [40] to visualize the 2D loss landscape, which is more easy to visualize the flatness of the loss landscape. As shown in Figure 12, the 2D loss landscape of our HUG loss is flatter than the widely used CE loss, showing that HUG yields a flat minima which may have better generalization ability.

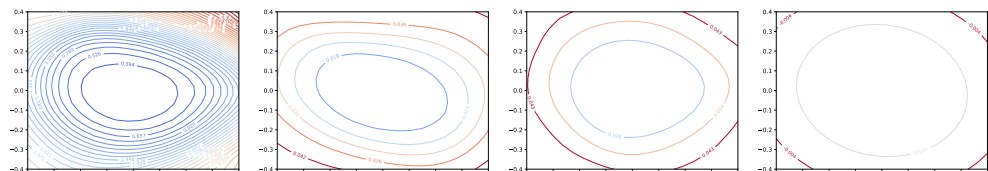

Figure 12: The 2D Loss Contour of different loss objective. From left to the right: (1). CE loss. (2). HUG overall loss. (3). intra-class loss. (4). inter-class loss.

**The ablation of $\alpha$ and $\beta$.** In our HUG framework, we introduce two scaling hyperparameters, $\alpha$ for the inter-class hyperspherical uniformity, $\beta$ for the intra-class hyperspherical uniformity. We investigate the effect of the two hyperparameters for the model performance. As shown in Table 8, HUG is not sensitive to $\alpha$, as the inter-class hyperspherical uniformity is always easy to optimize. HUG is also not sensitive to $\beta$ in a wide range. The ablations are conducted on CIFAR-100. $\alpha$ is set as 0.15 when we perform ablation on $\beta$. $\beta$ is set as 0.015 when doing ablation on $\alpha$.

| $\alpha$ | 0.0003 | 0.0015 | 0.015 | 0.05 | 0.15 | 0.5 | 1.5 | 5.0 |
|---|---|---|---|---|---|---|---|---|
| Accuracy | 76.31 | 75.99 | 76.28 | 76.16 | 76.48 | 76.32 | 76.1 | 76.03 |

| $\beta$ | 0.005 | 0.015 | 0.05 | 0.15 | 0.3 | 0.5 | 1.5 | 5.0 |
|---|---|---|---|---|---|---|---|---|
| Accuracy | 74.15 | 76.48 | 76.12 | 75.87 | 75.59 | 75.24 | 74.81 | 74.00 |

Table 8: Effect of hyperparameters $\alpha$ and $\beta$.

