# OpenReview forum: "Generalizing and Decoupling Neural Collapse via Hyperspherical Uniformity Gap"
_ICLR.cc/2023/Conference — ICLR 2023 poster_

### Official Review · Reviewer_bush · 2022-10-25

**Confidence:** 3
**Correctness:** 3
**Technical Novelty And Significance:** 3
**Empirical Novelty And Significance:** 3
**Recommendation:** 8

**Clarity, Quality, Novelty And Reproducibility:**

In my opinion, the work is of high-quality in that it provides sound theoretical foundations as well as convincing experimental results. Just some minor clarity issues:

- In Theorem 3, what does $\hat \mu_C^C$ mean when $C\to\infty$? I am confused since the $C$'s appear in both the subscript and the superscript.
- Also, $\hat\mu$ is not introduced until the line after Eq. (2), which is a projected vector on the unit-sphere. Is this the correct understanding? If it is, then this should be clarified in the paper.

**Strength And Weaknesses:**

Strength:
- The writing is good.

- The decoupling and the generalization of NC is excited to me, especially the solid theoretical analysis along with the great numerical performance of the proposed HUG loss (though I didn't check the proofs).

Weakness:
- It mentions that HUG serves as an alternative loss function in place of CE and MSE, but HUG is only compared with CE loss in the experiments while MSE is missing. Moreover, the following two work closely related to MSE loss and the NC phenomenon are missing in the paper.

    > Zhou, Jinxin, et al. "On the Optimization Landscape of Neural Collapse under MSE Loss: Global Optimality with Unconstrained Features." arXiv preprint arXiv:2203.01238 (2022).

    > Zhou, Jinxin, et al. "Are All Losses Created Equal: A Neural Collapse Perspective." arXiv preprint arXiv:2210.02192 (2022).

- The idea of GNC is great, but it is still under the assumption of a balanced dataset where each class has the same number of samples, which seems to be a bit conflict with the "generalized" claim. I think some comments and discussions are needed for this point.

**Summary Of The Paper:**

The paper analyzes neural collapse (NC) phenomenon by decoupling it into two separate learning objectives: minimal intra-class variability and maximal inter-class separability, and proposes a Generalized NC applicable to broader applications. It then designs a quality as  hyperspherical uniformity gap (HUG) working as an alternative but meaningful loss as for CE and MSE. Theories and experiments on classification tasks show the effectiveness of the HUG loss.

**Summary Of The Review:**

Based on the solidity of the work, as discussed above, I recommend for an acceptation.

---

> ### Author Response · Authors · 2022-11-19
> **Response to Reviewer bush**
>
>
> We sincerely thank the reviewer for the encouraging and constructive comments. We are super excited that the reviewer thinks highly of our work and recognizes our novelty. We take every raised question seriously and hope that our response can clarify your concerns. We will be more than happy to address any additional concerns.
>
>
> **Q1: It mentions that HUG serves as an alternative loss function in place of CE and MSE, but HUG is only compared with CE loss in the experiments while MSE is missing.**
>
> A1: Great question. We add the comparison to MSE on both CIFAR-10 and CIFAR-100 with ResNet-18. The testing errors are given below. Note that we use cross-validation to search the best hyperparameters for MSE. We can observe that HUG-MHE can still outperform MSE. We will include the full results in the final version.
>
> ||CIFAR-10|CIFAR-100|
> |---|:---:|:---:|
> |CE|5.45|24.90|
> |MSE|5.74|25.38|
> |HUG-MHE|**5.03**|**23.50**|
>
>
> **Q2: Moreover, the following two works closely related to MSE loss and the NC phenomenon are missing in the paper.**
>
> A2: We are sorry for missing these two related works. They are indeed highly relevant. For the first work, it characterizes the global solution when $d<C-1$, which we have discussed in the response to Reviewer k2o7 (Q3). For the second work, it examines a few popular loss functions (e.g., CE, MSE, FL) from a neural collapse perspective and gives useful analyses. We will cite and discuss both works in our final version.
>
>
> **Q3: The idea of GNC is great, but it is still under the assumption of a balanced dataset where each class has the same number of samples, which seems to be a bit conflict with the "generalized" claim. I think some comments and discussions are needed for this point.**
>
> A4: Sorry for the potential confusion. For the “generalized” part, we refer to the case where $d<C-1$ which is not considered or studied in the original NC. We will explain it more clearly and include more discussions in the final version.
>
> The assumption of a balanced dataset inherits from the original NC, so GNC does not generalize beyond this assumption. However, there is a recent work called minority collapse [Exploring deep neural networks via layer-peeled model: Minority collapse in imbalanced training, PNAS 2021] that explicitly looks into the imbalance dataset problem. Although this is out of the scope of this paper, we will give an in-depth discussion of the issue in the appendix of the final version.
>
>
> **Q4: In Theorem 3, what does $\hat{\mu}_C^C$ mean when $C\rightarrow \infty$? I am confused since the $C$'s appear in both the subscript and the superscript.**
>
> A4: Sorry for the confusion and thanks for pointing this out. The superscript $C$ denotes the construction of a converging sequence of point configurations. This comes from how the theorem is proved. Specifically, the proof requires us to construct a converging sequence where each element is a point set and its index is denoted by the superscript $C$.
>
> However, after the reviewer pointed this out, we realize that this superscript may be hidden in the main paper for clarity and only defined for usage in the proof, since the underlying indication is the same. We will revise this part to make it more clear in the final version.
>
>
> **Q5: Also, $\hat{\mu}$ is not introduced until the line after Eq. (2), which is a projected vector on the unit-sphere. Is this the correct understanding? If it is, then this should be clarified in the paper.**
>
> A5: Sorry for the confusion and the reviewer’s understanding is correct. We will give it a proper definition before using it. We will make the changes in the final version.

---

### Official Review · Reviewer_bWKx · 2022-10-27

**Confidence:** 4
**Correctness:** 2
**Technical Novelty And Significance:** 2
**Empirical Novelty And Significance:** 2
**Recommendation:** 3

**Clarity, Quality, Novelty And Reproducibility:**

**Already known theorems**.

The study of symmetric point configurations and energy minimizers on spheres is a classic problem and thus the paper can build upon a large body of existing results; some of these have already been used inside the machine learning community. In the current form of the paper, it is not clear, which results are considered new by the authors and which ones are considered established.

- Theorem 1. This is a theorem about the energy-minimizing configurations of convex kernels. An identical version of this theorem can be found as Theorem 2.4.1 in the monograph Discrete Energy on Rectifiable Sets by Sergiy V. Borodachov [1].
- Theorem 2. Admittedly, in the appendix it is clarified, that this can be concluded from Cohn & Kumar, 2007 [2]. A previous proof can be found in Theorem 3 of Yudin, 1993 [3].
- Theorems 3. and 4. are well-known and can for example be found in [1] (also mentioned in the appendix).
- Proposition 1. In the appendix it is clarified, that proposition 1 assumes *normalized* gaussian vectors. Then, this is an immediate consequence of the spherical symmetry of the gaussian distribution.
- Propositions 2 and 3 are applications of the theorems to the specific loss functions. In this sense, they are new.
- Proposition 4. This is a standard bound on the cross entropy loss which appears as an intermediate result in many papers about neural collapse for the cross entropy loss. Due to its simplicity, I don't think a reference is necessary, but calling it a proposition might be an overstatement.
- Theorem 5. This is a corollary from theorem 3, obvious enough that no proof is included in the appendix.

**Relation to contrastive learning**

The idea that inter-classes separability and intra-class variability is central to contrastive learning and has been applied there succesfully. This should definitely be discussed in the paper. In fact, the hyperspherical uniformity gap (HUG) proposed in this paper can be understood as a contrastive loss function. While recent works (e.g. [4,5]), typically utilize a loss function with a softmax function such that the attraction and repulsion forces are not obvious, older works (e.g. [6] minimize precisely, what the authors call hyperspherical energy (in [6] with $s=-2$). Particularly relevant to this paper are [7] which shows neural collapse for $c\le d+1$ and [8] which studies the asymptotic behavior for $c\to \infty$, i.e. they imply a result analogous to Proposition 3 for contrastive learning.


**The occurence of generalized neural collapse is not proven.**

First, the paper is misleading, as suggests that prior works on neural collapse (NC) did not discuss the setting of $d< C-1$ and that therefore a generalization to GNC is required. This might be unintentional though. It is true, that theorems in prior works, typically require $d\ge C-1$. But this should not be confused with that these works did not predict maximum inter-classes separability and minimum intra-class variability in general. The problem is that if $d< C-1$, there is no unique notion of maximum inter-classes separability and characterizing the loss minimizers is much more difficult. Particularly, different loss functions most likely have different minimizers. For example, [8] remark that
>" The assumption $c \le d + 1$ [variable names changed] is crucial, as it is a necessary and sufficient condition for the existence of the regular simplex. [...] If it is violated, then the bounds derived in §3 still hold, but are not tight. Studying the loss minimizing configurations in this regime is much harder. Even for the related and more studied Thomson problem of minimizing the potential energy of K equally charged particles on the 2-dimensional sphere, the minimizers are only known for K ∈ {2, 3, 4, 5, 6, 12} (Borodachov et al., 2019)."

Thus, I was very surprised and excited, that the authors defined GNC (2) via a minimum of Coulomb energy. While the minimizing configurations in this setting are equally not known, there is a large corpus of work dedicated to studying this setting. If it could be shown that a standard loss function like cross entropy would lead to minimal Coulomb energy, this would be very interesting and pave the way for future analysis. Regrettably, this is not shown. Instead, the authors analyze different loss functions. But even for these loss functions, only results for the well known settings $C\le d+1$. $C=2d$ and $C\to \infty$ are presented, i.e. settings where the minimizers are universal in the sense that they apply to a wide range of loss functions.
In the setting $C> d+1$ I doubt that the different HUG variants satisfy property (2) of GNC, as even s-Riesz potentials can have different minimizers depending on the value of $s$.
Overall, I wonder, *if the setting $C> d+1$ is studied only asymptotically, then why do we even need a generalized definition of neural collapse.*

Regarding the empirical evidence on page 4.
Figure 1 is misleading, as the 2-dim case is special because there GNC (2) is easily characterized by the angles between adjacent classes being constant and GNC(2) can be achieved. For higher dimensions, such an easy characterization impossible.
Even if we would minimize the Coulomb energy for $d=3$, the experiment would be indecisive about whether GNC (2) is achieved or not. This is because there is a large number of local minima with approximately equal potential energy, cf. [1, Section 2.4]. Thus, GNC(2) cannot be proven with gradient based optimization, and, contrary to claimed in the paper the experiments do not "verify the correctness of GNC(2)".

**Empirical evaluation**.

Important information is missing from the empirical evaluation.
- It is claimed, that the performance gains of HUG over cross entropy are significant. Yet, there is no reported standard deviation.
- How where the hyperparameters selected? Potentially, hyperparameters for HUG are tuned carefully but not for cross entropy.



References
[1] Discrete Energy on Rectifiable Sets, Sergiy V. Borodachov, Springer Monographs in Mathematics, 2019, https://link.springer.com/book/10.1007/978-0-387-84808-2
[2] Henry Cohn and Abhinav Kumar, Universally optimal distribution of points on spheres, 2007, https://www.ams.org/journals/jams/2007-20-01/S0894-0347-06-00546-7/
[3] The minimum of potential energy of a System of point charges, V. A. Yudin, 1993, https://doi.org/10.1515/dma.1993.3.1.75
[4] Chen et al., A Simple Framework for Contrastive Learning of Visual Representations, ICML 2020, https://proceedings.mlr.press/v119/chen20j.html
[5] Tian et al., What Makes for Good Views for Contrastive Learning?, NeurIPS 2020, https://proceedings.neurips.cc/paper/2020/hash/4c2e5eaae9152079b9e95845750bb9ab-Abstract.html
[6] Hadsell et al., Dimensionality Reduction by Learning an Invariant Mapping, CVPR 2006, https://ieeexplore.ieee.org/document/1640964
[7] Wang & Isola, Understanding contrastive representation learning through alignment and uniformity on the hypersphere. ICML 2020, https://dl.acm.org/doi/10.5555/3524938.3525859
[8] Graf et al., Dissecting Supervised Contrastive Learning, ICML 2021, https://proceedings.mlr.press/v139/graf21a.html

**Strength And Weaknesses:**

**Strengths**
- The paper addresses neural collapse in the setting when the number of classes is larger than the feature dimension, a much more difficult problem.
- The empirical evaluation not only considers classification accuracy in a standard setting but considers additional aspects, such as long-tailed recognition or adversarial robustness.

**Weaknesses**
- Many of the presented theorems are essentially known.
- Central to the paper is the design of loss functions that encourage inter-classes separability and intra-class variability. This is essentially the goal of contrastive learning. Relevant literature is neither mentioned nor discussed.
- Actually, it is not proven that generalized neural collapse (GNC) occurs if the feature dimension is smaller than the number of classes.
- The empirical evaluation misses relevant information.


**Summary Of The Paper:**

The paper considers the asymptotic training dynamics when classification loss approaches zero.
It generalizes the neural collapse phenomenon to settings to models trained on large numbers of classes.
Further, it proposes loss functions that decouple inter-classes separability and intra-class variability.
Theoretical properties of such loss functions are derived and empirically the loss functions are shown to be preferable over the cross entropy loss.


**Summary Of The Review:**

The main weakness is the limited novelty of the paper and I recommend rejection.

---

> ### Author Response · Authors · 2022-11-19
> **Response to Reviewer bWKx (Part 3)**
>
>
> **Q3: Actually, it is not proven that generalized neural collapse (GNC) occurs if the feature dimension is smaller than the number of classes.**
>
> A3: Thanks for the comment. Our paper **never** claims that GNC is proven for CE loss. This is the reason why we call it the GNC hypothesis. However, we do think it will be an important future work to prove GNC for CE loss, but it is way beyond the scope of our paper, since our paper is not targeted as a theory paper. As an encouraging sign, we show strong empirical evidence that GNC will very likely hold in training neural networks (with class-balanced datasets), which is also partially observed in deep face recognition networks (say [ArcFace: Additive Angular Margin Loss for Deep Face Recognition, CVPR 2019]).
>
> For the necessity of GNC, it is well motivated by practical scenarios of deep face recognition and extreme classification. Therefore, we believe such a generalized hypothesis will be necessary and important to study. Moreover, it readily inspires many well-performing alternative loss functions, with HUG being a general framework to design one. We are happy to see that the reviewer agrees on the significance of such a GNC hypothesis. Moreover, GNC reduces to NC when $d>C-1$, which makes GNC itself already a complete description of end-phase feature/proxy distributions. Despite not being completely proven (in the case of $d<C-1$), we have shown empirical evidence supporting its correctness. We believe that throwing an interesting question (and hypothesis) could sometimes be more important, and our work is exactly doing this.
>
> We are a bit confused by the reviewer’s statement of “Figure 1 is misleading”. Figure 1 aims to provide an intuitive understanding of why NC should be generalized and why GNC is a valid hypothesis. To this end, directly learning features of dimension 2 or 3 is the best way one could do. As one can see from Figure 1, NC is surely invalid anymore, and thus, it immediately motivates us to generalize NC (which also echoes the previous concern from the reviewer saying that GNC is not necessary).
>
> Since we agree that GNC is necessary, now we show that GNC’s characterization of final representation / proxy distribution indeed makes sense. First of all, all our empirical study shows that neural networks are indeed minimizing the hyperspherical energy as shown in Figure 3. As the reviewer has mentioned, there could be many local minima caused by gradient-based optimization. We don’t think it incurs a problem, as long as the neural network is indeed minimizing towards the minimum energy. The same problem also happens in NC in the sense that in practical network training, the proxies are only approximately forming an equiangular tight frame due to the non-convexity of neural networks. In fact, proving CE converges to NC with neural network features remains an open problem. The same type of hardness also applies to GNC. However, it still does not affect its usefulness of being a guiding hypothesis (just like NC), since we can effectively design new well-performing loss functions based on it.
>
> Finally, we share the same excitement with the reviewer that GNC is an interesting and potentially impactful hypothesis. We believe that it could be of broad interest to the ML community and truly hope that the misunderstandings regarding GNC could be addressed.
>
> **Q4: The empirical evaluation misses relevant information.**
>
> A4: Thanks for pointing this out. In fact, for supervised learning, the standard deviation is usually quite small, since the only randomness comes from uniform mini-batch sampling, which is shared by all the methods. To address the reviewer’s concerns, we have run both the CE and HUG-MHE (as a demonstrative example) for 5 different random seeds. The average performance and standard deviation are reported as below. We will include the full results for all the tables in the final version.
>
> ||CIFAR-10|CIFAR-100|
> |---|:---:|:---:|
> |CE|5.497±0.219|24.985±0.328
> |HUG-MHE|**5.028±0.224**|**23.42±0.301**|
>
> For the hyperparameter selection, we use standard cross-validation which divide the original training set to two subsets (one for training and one for validation), and tune the hyperparameters there. We note that the CE loss does not have hyperparameters and the standard setting is already the best one (with typical momentum SGD, lr=0.1, momentum=0.9). We have varied these possible hyperparameters and CE already achieves the best performance.

---

> > ### Comment · Reviewer_bWKx · 2022-11-21
> > **Regarding the definition of GNC and that GNC is not proven.**
> >
> > Again, I did not write that the paper claims to prove GNC for the cross entropy loss. However, I pointed out, that in the submission, GNC is not proven for any loss function at all (disregarding the special case of NC and the asymptotic regime). Here, with GNC I specifically mean GNC (2) which is the non-obvious part. To be clear, with GNC(2) I refer to its explicit formulation in the paper, i.e., minimal Riesz-2 energy. This is different to what is observed empirically (and what is already known from prior work on neural collapse), i.e., that the distances between class centers become in some sense uniform and maximal. **GNC(2) is a far stronger definition, as it quantifies 'in some sense uniform and maximal'**.
> > I also want to point out, that GNC(2) is defined via minimal Riesz-2 energy and this is called minimal Coulomb energy. In physics, Coulomb energy is proportional to $r^{-1}$ or more generally, to $r^{d-1}$, instead of $r^{-2}$.
> >
> > In the response, authors state that "However, we do think it will be an important future work to prove GNC for CE loss". As I tried to communicate in my review, I would be very surprised if GNC(2) can be proven for the CE loss; I expect it to be disproved.; I expect that the loss minimizers differ depending on the loss function. This is because even for Riesz-s potentials the loss minimizers disagree depending on the choice of $s$. In the language of the paper, Riesz-2 is MHE-HUG, whereas MHS is related to Riesz-(-1). Mean squared error would be $s=-2$.
> > So far, there are more indications that the hypothesis of GNC(2) is false, than for it being true. In other words, **the hypothesis of GNC(2) is most likely false.**
> >
> > Consequently, I disagree with that the paper contains "strong empirical evidence that GNC will very likely hold in training neural networks". I was trying to make this clear in the paragraph which contains "Figure 1 is misleading", where I argue that the setting of Figure 1 (2-dimensional features) is the *only one*, where I would expect GNC(2) to hold. Admittedly, I agree that "one can see from Figure 1, NC is surely invalid anymore"[for 10 classes in \mathbb R^2]. Clearly, Figure 1 shows that an equiangular triangle cannot have more than 3 vertices.
> >
> > Last, I need to comment on the following paragraph
> > >As the reviewer has mentioned, there could be many local minima caused by gradient-based optimization. We don’t think it incurs a problem, as long as the neural network is indeed minimizing towards the minimum energy. The same problem also happens in NC in the sense that in practical network training, the proxies are only approximately forming an equiangular tight frame due to the non-convexity of neural networks. In fact, proving CE converges to NC with neural network features remains an open problem. The same type of hardness also applies to GNC.
> >
> > I disagree, that the same problems happen to NC and GNC. For NC, the loss minimizing configurations are known to be a simplex for typically used loss functions, eg. [8, 11], at least if one assumes unconstrained features. Further, the loss surface is in some sense benign (e.g. [12-14], such that GNC can be achieved with gradient based optimization. For GNC, already the loss minimizers might not be as hypothesized. Furthermore, for certain (easy) loss functions (e.g. Riesz energy), the loss surface is known to be malign. The non-convexity of neural networks is an additional obstacle unrelated to my critique.
> >
> > [12] Zhu et al., A Geometric Analysis of Neural Collapse with Unconstrained Features, NeurIPS 2021
> > [13] Han et al., Neural Collapse Under MSE Loss: Proximity to and Dynamics on the Central Path, ICLR 2022
> > [14] Yaras et al., Neural Collapse with Normalized Features: A Geometric Analysis over the Riemannian Manifold, https://arxiv.org/abs/2209.09211

---

> > > ### Author Response · Authors · 2022-11-21
> > > **Thanks for the comments!**
> > >
> > > We are deeply appreciative of the reviewer's comments. They are super helpful. With all due respect, we think the reviewer may misunderstand our paper in a few aspects.
> > >
> > > To start with, despite some disagreements, we do agree that GNC can indeed be written in a more general form by turning Riesz-2 energy to a Riesz-s energy, which can be easily revised in our final version. This will resolve most of the reviewer's concerns on $s$. Most importnatly, theorem 1 already applies to riesz-s energy, so there is very little thing to change. In fact, s-energy is actually the intention of GNC, since the generalization mostly comes from hyperspherical energy which is defined by Riesz-s energy. Thanks a lot for the great insight! Super helpful!
> > >
> > > First, with all due rescpect, we disagree that GNC(2) is a far stronger definition. It really depends on what you are comparing to. If you are comparing to NC(2), then GNC(2) is definitely a weaker definition since it generalizes NC(2) and make it a special case. If you are comparing to the other form of energies, then making the definition s-energy could address this. It does not affect our central idea to generalize NC to GNC with hyperspherical energy.
> > >
> > > Second, with all due rescpect, we disagree that the statement that the hypothesis of GNC(2) is most likely false. There is no evidence at all that GNC is invalid (both theoretically and empirically). **Since the reviewer did not raise any factual evidence it can be disapproved, we think GNC is still a promising way to generalize NC.** Esepcially we agree upon that NC should be generalized, and our paper makes one of the earliest attempt in this direction.
> > >
> > > For the last point, the reviewer may misunderstand what GNC(2) really means. GNC(2) is characterized by the minimizer of the hyperspherical energy that effectively generalizes simplex ETF. The whole point of GNC is that it does not contradict with NC, yet still being able to cover the case of $d<C-1$. GNC also provides a sensible hypothesis for this case, since all the empirical studies show that CE empirically converges to GNC. **We want to re-emphasize that GNC(2) is characterized by the minimizer of the hyperspherical energy, rather than the minimization itself. Therefore, the loss surface of minimizing hyperspherical energy is not relevant here and is also not the concern of GNC.** We simply use a variational characterization to define GNC, so whether the loss surface of hyperspherical energy minimization does not matter to GNC at all. Maybe the reviewer misunderstood this part?
> > >
> > > But we agree with the reviewer that proving CE converge to GNC is a difficult problem. However, NC and the asymptotical case ($N\rightarrow \infty$) are very positive signals that GNC could be true. We argue that GNC surely provides an interesting and impactful theoretical problem for future study. Moreover, the difficulty mostly comes from the problem setting, i.e., the relationship between feature dimension and number of classes, rather than GNC itself. In summary, we believe that the characterization of GNC is valid and useful.
> > >
> > > On the other hand, the implication of GNC could be even more imporant than whether it can be proved or not. As one can see, the design of HUG is guided by GNC, and its performance is better than CE in many cases. It already validates the power of GNC and demonstrates why GNC is very important and useful as a guiding hypothesis / principle.
> > >
> > > Finally, we want to sincerely thank the reviewer again. Any additional comments or conerns are extremely welcome. We confirmly believe that the reviewer's comments are very helpful to improve our paper.

---

> ### Author Response · Authors · 2022-11-19
> **Response to Reviewer bWKx (Part 2)**
>
>
> **Q2: Central to the paper is the design of loss functions that encourage inter-classes separability and intra-class variability. This is essentially the goal of contrastive learning. Relevant literature is neither mentioned nor discussed.**
>
> A2: Thanks for the comment. While we acknowledge the connection to contrastive learning, we, with all due respect, disagree that the goal of contrastive learning is the same as standard supervised learning which is the setting we consider.
>
> First of all, we would like to highlight that there is essentially no concept of classes in contrastive learning, and consequently, no concept of proxies (i.e., last-layer classifiers). This makes contrastive learning and standard supervised learning very different. Contrastive learning is typically considered as proxy-free pair-wise learning where either positive or negative pairs are presented in one shot and features are directly compared without an intermedia proxy, while multi-class supervised learning is typically proxy-based triplet-wise learning where both positive and negative pairs are presented in one shot (through an anchor sample) and proxies are used to represent a group of features. Please refer to [Circle Loss: A Unified Perspective of Pair Similarity Optimization, CVPR 2020] and [SphereFace2: Binary Classification is All You Need for Deep Face Recognition, ICLR 2022] for a complete comparison.
>
> Moreover, our inspiration is drawn from the classical Fisher discriminative analysis where maximal inter-class separation and minimal intra-class variability are imposed. In modern representation learning, these two criteria were explicitly considered in deep face recognition (see [SphereFace: Deep Hypersphere Embedding for Face Recognition, CVPR 2017], [Cosface: Large margin cosine loss for deep face recognition, CVPR 2018], [ArcFace: Additive Angular Margin Loss for Deep Face Recognition, CVPR 2019] and many works mentioned therein). In fact, our setting of $d<C-1$ is exactly from deep face recognition where one typically has a million-class datasets and has to resort to a much smaller feature dimension (typically 512).
>
> Second, our work is mostly aligned with and inspired from the series of works in neural collapse where contrastive learning is typically not viewed as an example, for example, see [Neural Collapse Under MSE Loss: Proximity to and Dynamics on the Central Path, ICLR 2022] and [Prevalence of neural collapse during the terminal phase of deep learning training, PNAS 2020]. Neural collapse focuses on the training dynamics and convergent representation behaviors for standard CE loss and MSE loss, both of which couple inter-class separability and intra-class variability. In contrast, HUG decouples these two criteria with a novel formulation from the characterization of hyperspherical uniformity. We show that HUG is able to readily serve as an alternative supervised learning objective to CE loss. We argue that sharing a high-level goal of optimizing these two criteria (like CE and MSE) does not affect the novelty of the HUG objective.
>
> Third and more specifically, contrastive learning and supervised learning consider different notions of uniformity. For contrastive learning, because there is no concept of classes, uniformity refers to the sample-wise uniformity, while ours considers class-wise uniformity. These two targets are totally different. some recent works ([Understanding the behaviour of contrastive loss, CVPR 2021] and [Your Contrastive Learning Is Secretly Doing Stochastic Neighbor Embedding, NeurIPS 2022]) argued that extreme sample-wise uniformity in contrastive learning would damage the latent semantic structure and could harm out-of-distribution generalization, which is called “Uniformity-Tolerance Dilemma”. In contrast, we consider supervised learning where inter-class separability generally will not damage the latent semantic structure as well as generalizability. Moreover, there is also no intra-class variability in contrastive learning, since it promotes instance-level discrimination.
>
> However, we definitely agree with the reviewer that there are connections between contrastive learning and supervised learning, especially when they are applied on the same class-based datasets. We are appreciative of the reviewer for mentioning all these highly relevant and inspiring works. We will definitely cite and discuss the mentioned papers in the final version.

---

> > ### Comment · Reviewer_bWKx · 2022-11-21
> > **Connections to contrastive learning.**
> >
> > I did not state that "the goal of contrastive learning is the same as [the one of] standard supervised learning". What I wrote was promoting inter-class separability and intra-class variability is central to contrastive learning. I should have been more clear, though. In the typical semi-supervised setting of contrastive learning, separability, resp. variability, is not promoted with respect to classes, but with repsect to samples/views. However, contrastive learning has also been applied successfully in supervised settings, see for example [9,10].
> >
> > Neglecting the above, the goals (attracting similar samples / classes, repulsing dissimilar samples / classes)  are essentially the same. Therefore, I disagree with "Third and more specifically, contrastive learning and supervised learning consider different notions of uniformity." Mathematically, the notions of uniformity are the same, they only differ in the regarded quantities.
> > Thus it is not surprising, that parts of the analysis in this paper have already been done for typical contrastive loss functions. As I said, the limit of classes to $\infty$ (in the reference, this corresponds to the number of negative samples) has already been studied in [7] and the setting of less classes than dimensions has been studied in [8] (this reference considers the supervised setting). Just as [11] proves Proposition 3 for cross entropy,  in their combination, both references [7,8] imply Proposition 3 for the (supervised) contrastive loss function. I did not see the latter in the updated version.
> >
> > References:
> > [9]  Khosla et al., Supervised Contrastive Learning, Neurips 2020,
> > [10] Gunel et al.,  Supervised contrastive learning for pre-trained language model fine-tuning, ICLR 2021
> > [11] Lu and Steinerberger, Neural collapse with cross-entropy loss, https://arxiv.org/abs/2012.08465

---

> > > ### Author Response · Authors · 2022-11-21
> > > **Thanks for the comments!**
> > >
> > > We are deeply appreciative of the reviewer's comments. With all due respect, we think the reviewer may misunderstand our paper in a few aspects.
> > >
> > > For the goal of contrastive learning, the purpose of our rebuttal is to differentiate their use case. Our use case is in general classification, where ce loss is usually applied. Contrastive learning considers a quite different setting for discriminating instances. There are many scenarios where ce loss / HUG is appliable, but contrastive leanring is not. For example, for the data that is highly nontrivial to augment, say molecules, graphs, tabular data and text, how to apply contrastive learning is still an open problem. On the other hand, both CE loss and HUG consider the case where the labels exist. To reiterate, the purpose of our paper is to work on a new framework to design losses that are alterantive to CE. Therefore, the connection to contrastive learning does not affect our novelty and contributions.
> > >
> > > Even if contrastive leanring is studied in a supervised case (as the reviewer mentioned), the settings are still very different, since contrastive learning does not consider proxies. As a concrete example, contrastive learning typically needs to learn proxies afterward or use K-NN classifiers, while CE or HUG learns classifiers jointly.
> > >
> > > For the notion of uniformity, our rebuttal emphasizes that contrastive learning and supervised learning do not converge to the same uniformity notion. Class uniformity is a much more relaxed requirement than sample uniformity, and class uniformity is subject to human intervention from labeling. However, for the sample uniformity, it is a very strong constraint and it can even hurt the generalization performance. Contrastive learning is more like exploring the intrinsic structures hidden in the data and take advantage of it in learning representations, while supervised learning is simply to predict the label from the observation. Moreover, mathematically, we use more characterizations than contrastive learning (say MHS and MGD).
> > >
> > > In general, we definitely agree with the reviewer that the connection to contrative learning should be discussed and we will do it properly in the final version. However, we want to emphasize that our paper is mostly working on understanding, generalizing and decoupling neural collapse. This is a very different direction than contrastive leanring.
> > >
> > > Therefore, it seems to be a bit farfetched for our paper to focus significantly on contrastive learning which is not a main theme in neural collapase (which can be seen from the references in all the related works about neural collapse).

---

> ### Author Response · Authors · 2022-11-19
> **Response to Reviewer bWKx (Part 1)**
>
>
> We deeply appreciate the efforts and time that the reviewer put in our paper. All the detailed comments and suggestions (including the criticism) are extremely useful for improving our paper. However, with all due respect, there are many major misunderstandings regarding the motivation, contribution and novelty of our paper. Please bear with us for a relatively long response. Hopefully, our rebuttal can clarify such misunderstandings and address the reviewer’s concerns. We will be more than happy to address any additional concerns.
>
> **Q1: Many of the presented theorems are essentially known.**
>
> A1: Thanks for the detailed comments. First of all, we have **never** claimed that these theorems are completely our contributions. The reviewer may refer to our list of contributions at Page 2 of our main paper, none of which has taken these theoretical results as our own contributions. However, rather than the proving techniques for these theorems, our contributions lie in connecting the active research area of neural collapse with the classical potential theory (e.g., minimizing discrete Riesz energy), and building deeper insights into the behavior of neural networks. Such a connection is actually very important in the sense that many highly nontrivial and useful theoretical results from potential theory can be adapted here to understand the training of neural networks. This is exactly the reason that we hypothesize generalized neural collapse (to connect neural networks with potential theory).
>
> Second, we also want to point out that the significance of a theoretical result may not be judged from whether it is easy to prove, but rather how it can help us gain deeper insights / understandings. And for this reason, our section is termed “theoretical insights” instead of “theoretical guarantees”. For example, the proposition that normalized zero-mean gaussian leads to uniform distribution on the hypersphere is easy to prove, but it induces interesting understandings for why neural networks are initialized with low hyperspherical energy. This is because Xavier and Kaiming initialization are all zero-mean gaussians. This also helps us understand that hyperspherical uniformity among proxies of different classes is guaranteed to be small from the very beginning, and based on this observation, we propose that fixing randomly initialized proxies is a sensible approach for training the proxies (which is verified by our “static proxies” proposal).
>
> To summarize, we want to emphasize that our contribution is more about the generalized neural collapse hypothesis which effectively connects neural network training and potential theory (as in comparison, neural collapse connects neural network training and tight frame theory). More importantly, we propose a new framework for designing loss functions that is motivated by GNC but with additional flexibility from decoupling GNC (unlike CE and MSE).
>
> Last but not the least, we also want to thank the reviewer for the suggestion for being more clear about what are proven results. We will revise the paper accordingly in the final version.

---

> > ### Comment · Reviewer_bWKx · 2022-11-21
> > **Regarding the theorems.**
> >
> > I admit, the theorems are not listed as a contribution on page 2. Still, the theorems are stated without referring to their origin. This suggests to the reader that these results are novel. Further, it does not give proper credit to the originators of the results. This issue could easily have been fixed in the updated version (I even listed references in my review and some references can already be found in the appendix), but it wasn't.
> >
> > Further, I never said that I judge the significance of the theorems on how difficult they are to proof. Instead I expressed that the theorems (except Propositions 2 and 3) are not novel and considered this aspect when evaluating the theoretical contribution of the paper.
> > As the authors clarified now (in Part 3), the submission is not intended as a theory paper. Considering the 'theorem density' in the paper, this was not obvious.

---

> > > ### Author Response · Authors · 2022-11-21
> > > **Thanks for the comments!**
> > >
> > > We are deeply appreciative for the reviewer's comments.
> > >
> > > First of all, we indeed aim to refine the presentation of the theorems. Given there are some other suggestions from the other reviewers (e.g., mention experimental results earlier in the paper, and more discussions to some related work) and the very short amount of time for rebuttal (for running additional experiments, preparing text rebuttal, etc.), we are unable to revise the paper immediately to a final version. However, as we promised in the summary, we will make all the revisions mentioned in our rebuttal in the final version. We are sorry that the current version may not be able to reflect all the intended changes.
> > >
> > > Second, we feel like the reviewer has some misunderstandings on the usage of the theorems. They are intended to be "theoretical insights", as we stated in the section title. They can help use understand how GNC is connected to potential theory. But we do agree with the reviewer that the presentation itself can be improved.

---

> ### Author Response · Authors · 2022-11-22
> **Summary of Round-2 Rebuttal**
>
> We sincerely thank the reviewer for spending time in reading our rebuttal and giving constructive feedback. We are glad that partial concerns of the reviewer (Q4) has been addressed. Despite disagreement on some issues, the reviewer’s suggestions are extremely important to improve our paper. We deeply appreciate it.
>
> To ensure an effective communication, we briefly summarize the reviewer's concerns and how we addressed them.
>
> **Q1. The presentation of theoretical results**
>
> A1. Our theoretical results are used to explain how GNC is related to NC, and more broadly, classical potential theory. To reiterate, our paper aims to build connections between GNC and many nice results in potential theory, rather than working on new results in potential theory. We believe that the connection to NC and potential theory is novel and GNC is of great significance to the study of neural collapse. Due to the short amount of time for preparing rebuttal, we can only incorporate partial revision to the current version. As we stated in the summary, we will make all the changes promised in the rebuttal in the final version.
>
> **Q2. The goal of contrastive learning**
>
> A2. What we are trying to convey is that contrastive learning and supervised learning are solving different problems and require different formulations for the objective functions. What our paper is focusing on is the standard supervised learning where CE or MSE loss are typically studied. With all due respect, we disagree with the reviewer that “the goals (attracting similar samples / classes, repulsing dissimilar samples / classes) are essentially the same”.
>
> First of all, we stick to our previous statement that sample discrimination and class discrimination are very different. Sample discrimination is more like a limiting case for class discrimination (the number of class is equal to the number of samples), and they don't usually converge to the same solution.
>
> Second, our ultimate goal is to generalize and decouple neural collapse such that a new family of loss functions can be inspired. Neural collapse is to study the training dynamics and how representations converge to, whose focus is very different with contrastive learning. More importantly, the connection to contrastive learning does not affect our contributions. We also want to emphasize that our work is aligned with neural collapse, rather than contrastive learning.
>
> However, the references the reviewer provided are definitely relevant, we will cite and discuss them in our final version.
>
> **Q3: Whether GNC is valid**
>
> A3: First of all, GNC, as a hypothesis, is not disapproved of by any factual results (either theoretical or empirical). There is also no evidence suggesting that GNC is invalid. On the contrary, there is much empirical evidence that shows that proxies / class-means are approximately converging to GNC.
>
> To clarify, GNC uses hyperspherical uniformity to generalize NC, making NC a special case. Such a hyperspherical uniformity is characterized by the minimizer of hyperspherical energy which covers the case of simplex ETF. The nonconvex minimization of hyperspherical energy is not relevant in GNC. Neural collapse becomes much more complex when $d<C-1$. It is the problem nature that complicates the loss landscape of CE, not the hypothesis of GNC. We believe that GNC is still one of the cleanest yet intuitively sensible ways to generalize NC. As one of the earliest attempts to generalize NC, we believe GNC is of great significance and it is also novel.
>
> Finally, we believe that the implication and usefulness of GNC is even more important than whether it can be proved or not. The fact that GNC inspires the HUG framework is already a good example, in the sense that HUG can easily outperform CE. We feel that GNC can inspire many more interesting follow-up not only for understanding neural collapse but also for designing well-performing loss functions.

---

> ### Author Response · Authors · 2022-11-22
> **Additional information to help clarify the concerns**
>
> In the round-2 rebuttal summary, we have summarized all the three remaining concerns from the reviewer. We would like to give additional responses to help better clarify the reviewer’s concerns as well as potential misunderstandings.
>
> ======================= Q1 =======================
>
> For Q1 (the presentation of theoretical results), we think we are on the same page with the reviewer that the presentation of theorems can be improved, and that the contribution of connecting GNC (and NC) with potential theory is novel and useful. As the reviewer wrote, this concern can be addressed with some simple revisions. We definitely agree on this and will keep revising the paper for the final version.
>
> ======================= Q2 =======================
>
> For Q2 (the goal of contrastive learning), we hope that the reviewer agrees with us that sample-wise uniformity and class-wise uniformity is different. A very simple example will be that labels can be randomly assigned and still learnable, while contrastive learning is not designed to do this. Nonetheless, all we are trying to say is that CE / HUG has different use cases with contrastive learning. Despite the fact that contrastive learning can be studied in a supervised setting, it does not change the fact that they are different. More importantly, existing works on neural collapse do not view contrastive learning as an example, which is the reason why we didn’t think of contrastive learning in the first place. With that being said, **we feel that it may be a bit unfair for our paper that the connection to contrastive learning becomes a weakness. Instead, it should be an interesting and additional discussion that connects neural collapse and contrastive learning**.
>
> ======================= Q3 =======================
>
> For Q3 (whether GNC is valid), since there is no factual evidence supporting that GNC is invalid and all the empirical results we have show that GNC is approximately attained, there is no reason to think that GNC is likely false. The only facts we know about GNC is that **(1) GNC makes NC and the asymptotic case ($N\rightarrow\infty$) special cases and is theoretically correct in these two cases; and (2) for the case of $d<C-1$, all the current empirical experiments do not contradict with the hypothesis of GNC.**  Therefore, we have good reason to believe GNC will hold (at least it will be useful) as a hypothesis.
>
> To summarize, **the central idea of GNC is to generalize simplex ETF in NC to hyperspherical uniformity, and hyperspherical energy is simply one variational way to characterize hyperspherical uniformity.** We do think that there could be some other alternative (potentially better) characterizations. But the current hyperspherical energy is the best and only characterization we could think of for now, and more importantly, it serves as an important bridge to connect all the nice theoretical results/insights from potential theory. We believe these advantages already suffice to make the hyperspherical energy characterization significant and useful enough.
>
> Most importantly, as we emphasize previously, the usefulness and significance of the GNC hypothesis is not simply about how to prove it, but rather whether it is inspiring and useful as a guiding principle for future research. GNC directly inspires the HUG framework for designing loss functions that are alternative to and perform better than the CE loss. We believe there could likely be more follow-up works for new well-performing loss functions.
>
> ======================= Summary =======================
>
> To the three remaining concerns, we feel like Q1 and Q2 are generally very minor issues that can be easily addressed in our final version. For Q3, it is more about how one understands the problem and how one positions GNC. GNC, as a hypothesis, already makes one of the earliest yet necessary steps to generalize NC to cover all the practical application settings (say deep face recognition). The hyperspherical energy minimizer characterization is not only theoretically sound (it provably unifies NC and the asymptotic case), but also empirically supported (it gives empirically valid conjecture on how class-means distribute in the case of $d<C-1$). We don’t aim to conclude that ours is the optimal or the best characterization, but it is indeed an important step forward from simplex ETF. We sincerely hope that the concerns from the reviewer can be better addressed by now.
>
> Finally we want to express our deep gratitude to the reviewer for all the time and effort made to give valuable comments. Free feel to let us know if there are any additional suggestions or concerns. We are more than happy to address them.

---

> > ### Comment · Reviewer_bWKx · 2022-11-22
> > **Response to authors**
> >
> > Dear authors,
> >
> > I will keep it short.
> >
> > **Q1**. Please include references for all theoretical from the literature (it seems we agree in this aspect).
> >
> > **Q2**. Please include a short paragraph on the conceptional similarity (promoting attraction and repulsion) to contrastive learning. In the proximity of Proposition 3, state that Proposition 3 has been proven for the cross entropy loss [11] and for (supervised) contrastive loss [7,8].
> >
> > **Q3**. At this point I have come to terms with that you cannot be convinced that GNC is a unlikely hypothesis and that the paper does not provide sufficient evidence. I will raise one last issue. Afterwards I will not comment further on this aspect.
> > *Maximum hyperspherical separation (MHS) does not imply GNC.* To recall, maximum hyperspherical separation  is defined via $\max_{ \hat w_i } $ $\min_{i\neq j} \lVert \hat w_i -\hat w_j \rVert$. So MHS is just another name for the best packing problem.
> >  The maximizer of MHS is not unique because of the minimum operator in the definition, while the minimizer of Riesz-1 energy is unique (up to isometries). For example, for 5 points on the 2-sphere, a best packing solution is achieved if $w_1$ and $w_2$ are antipodal and $w_3, w_4, w_5$ are on the corresponding equator and have pairwise (geodesic) distance $\ge \pi/4$  (cf. [1, Theorem 3.2.2]). Thus maximizing MHS will stop with such a configuration. But changing the positions $w_3, w_4, w_5$ on this equator will still change the Riesz-1 energy and GNC(2) is only achieved if $w_3, w_4, w_5$ form a equiangular triangle [15].
> >
> > [15] Schwartz, The Five-Electron Case of Thomson’s Problem, https://doi.org/10.1080/10586458.2013.766570

---

> > > ### Author Response · Authors · 2022-11-23
> > > **Thanks for the comments**
> > >
> > > We are very glad to see that the major concerns regarding Q1, Q2 and Q3 have been addressed. We are deeply thankful for the vivid discussion that improves our paper. We are also more than happy to address any other concerns.
> > >
> > > For the new issue raised by the reivewer, we agree with the reviewer that 2-energy MHE and MHS are not equivalent (since $s$ are different in two cases), and we are aware of this result. We are also aware of the 5-electron result mentioned by the reviewer, and this result does not contradict the current claims in our paper. **We didn't claim that MHS attains 2-energy GNC in our paper**. We note that, the core idea of GNC is to replace simplex ETF with hyperspherical uniformity where a number of variational characterizations are available. MHS is simple one of them and serves as a heuristic under the HUG framework.
> > >
> > > Most importantly, as we mentioned in one of the previous responses, GNC can be slightly generalized with $s$-energy (which is exactly the formulation used in MHE). **Since MHS can be viewed as a special (limiting) case of MHE, as $s$ goes to infinity, MHS naturally becomes part of the $s$-energy GNC hypothesis.** This makes GNC more flexible for different chariacterizations. We will surely apply this change to our final version.
> > >
> > > As we emphasized in our previous response, there are definitely many ways to characterize hyperpsherical uniformity (there are even definitions for hyperspherical quasi-uniformity or probabilistic uniformity). But **the key contribution of GNC is to make one of the earliest effrots for generalizing from simplex ETF to hyperspherical uniformity.** The idea itself is more important and inspiring than the specific characterization. The reason we seek to use minimal hyperspherical $s$-energy characterization is that it naturally unifies many interesting characterizations already, for example, best packing (MHS), and connects directly to potential theory (where many nice theoretical results are available).
> > >
> > > At the end of the day, we definitely don't want to claim that minimal $s$-energy characterization is the best way to formulate GNC, but it is definitely interesting and useful since it connects directly to potential theory, and it is also sufficiently general to cover a number of formulations. We expect that our paper can be an inspiration for many other (potentially better) hyperspherical uniformity characterizations.
> > >
> > > Finally, we would like to thank the reviewer again for the constructive comments. We deeply appreciate it.

---

> ### Author Response · Authors · 2022-11-23
> **Summary of Round-3 Rebuttal**
>
> ======================= Summary =======================
>
> We sincerely thank the reviewer for the vivid discusssions. We are grateful for all the time that the reviewer spent in our paper. **To summarize, we are very glad that most concerns regarding Q1, Q2 and Q3 are addressed.** Till now, it seems that there are no major concerns for the motivation, related works, contribution, novelty and experiments regarding our paper. If there are any other concerns, we are surely happy to address them (both verbally and experimentally).
>
> ======================= the newly raised question =======================
>
> We are definitely happy to see this question raised by the reviewer, since it is also related to why GNC is an important idea. MHS (best packing on sphere) shares the same simplex ETF solutions with 2-energy MHE when $d>C-1$, so both of them can properly generalize NC. The difference is that how they behave in the case of $d<C-1$. The 5-electron case indeed shows that they may have different minimizers (as indicated by section 2.5 in the discrete energy book). Therefore, we believe that a proper generalization of NC will be necessary and important, and also remains an open problem. GNC simply makes the first effort towards this end.
>
> To directly answer the reviewer's question, a $s$-energy formulatiom for GNC can well ressolve this issues while making GNC even more general. Despite this, we want to emphasize that **the core idea of GNC is to generalize simplex ETF to hyperspherical uniformity**. As a wild guess, the specific $s$ in GNC may likely depend on properties regarding the neural encoder (so-called inductive biases). Although it is beyond the scope of our paper, this could be another interesting future work.
>
> To conclude, we believe that this issue not only highlights the importance of GNC but also reveals many potential future works. In this regard, **our paper formally posts this question of generalizing NC (with a generali idea of hyperspherical uniformity, a specific energy characterization and an resulting loss design framework)**. We think our paper will be a significant addition to neural collapse, and will also be inspiring for understanding the behavior of neural networks.

---

### Official Review · Reviewer_k2o7 · 2022-10-29

**Confidence:** 4
**Correctness:** 3
**Technical Novelty And Significance:** 3
**Empirical Novelty And Significance:** 3
**Recommendation:** 8

**Clarity, Quality, Novelty And Reproducibility:**

**Clarity**
The definitions and theoretical statements are clear. However, the exposition can be improved (HUG is not introduced until page 5), some discussions of prior works are missing, and some implementation details are unclear.

**Novelty**
The HUG objective, the experiments, and their associated theoretical analysis are definitely novel ideas. However, similar ideas to those discussed by the authors *have* appeared in the literature (such as Lu and Steinerberger and Zhou et al. 2022).

**Reproducibility**
* There is no reproducible code provided with this paper.

**Strength And Weaknesses:**

**Strengths**
 The authors show theoretical derivations and discussions as well as numerical experiments supporting the use of HUG. The theorems and experiments, for the most part, are concise and easy to understand---supporting the authors' arguments. The appendix provides additional thought-provoking experiments and ablation studies.

**Weaknesses**
This paper has notable weaknesses in its clarity and discussion of prior works.

I describe potential improvements below:

**Major Comments**
* The authors do not provide any reproducible code. Might is be possible for the authors to give short, self-contained code that (at least) shows the implementation of MHE, MHS, and MGD?
* Lu and Steinerberger [39] also discusses the connection of hypersphere uniformity with neural collapse in their Section 1.4. Moreover, they discuss this relative to a "frame potential" along the same lines as to the energy minimization discussed by the authors of this paper's theory. I realize the authors already briefly acknowledge [39], but is done only in one passing sentence. Given the similarities in these two works, might the authors add a slightly longer discussion/comparison of these two works?
* The following ICML paper also examines neural collapse when the number of classes is larger than the ambient dimension:

  Zhou, J., Li, X., Ding, T., You, C., Qu, Q. &amp; Zhu, Z.. (2022). On the Optimization Landscape of Neural Collapse under MSE Loss: Global Optimality with Unconstrained Features. Proceedings of the 39th International Conference on Machine Learning in *Proceedings of Machine Learning Research* 162:27179-27202. https://proceedings.mlr.press/v162/zhou22c.html.

  Might the authors include additional discussions on how their GNC definition compares to the generalization of NC in the $d < C-1$ case given by Zhou et al. 2022?

* Page 6-7: The implementations of MHE, MHS, and MGD all require the computation of *all pairwise distance norms* for the vectors passed to the associated HU(...) function. Computationally, this seems significantly slower than MSE or CE which do not require a quadratically-sized outer loop. Might the authors comment on this?

*  On the same note: Might the authors discuss or include an experiment comparing the compute/CPU time required to train using HUG vs. using CE or MSE?

**Minor Comments**
* The authors discuss HUG extensively in pages 1-4, but HUG is not introduced until Page 5. As a result, for four entire pages, the reader is left to purely imagine what HUG (hypothetically) might be. This significantly weakens the authors' discussion. I strongly recommend introducing the formal definition of HUG earlier.
* Page 2: Are Projection FDA and Data FDA instances of HUG with a particular choice of HU(...)? This is especially unclear at this point of this paper since HUG has not been introduced yet.
* Page 2: It seems there is an even bigger problem with Projection FDA and DataFDA as objectives: They require explicitly calculating the Sb and Sw of (at best) the minibatch and (at worse) the entire dataset. This is very memory and computationally expensive. The inverse will also be expensive. MSE and CE training do not require these matrix quantities and would be much more efficient. Might the authors comment on this?
* Page 5: "Nontrivial to optimize the original HUG..." At this point, the reader has not seen a concrete example of the HU(...) function yet. Thus, it is unclear why the original HUG is difficult to optimize. Even after seeing concrete examples in later papers, I am not sure which particular difficulties the authors are referring to.
* Page 8-9: I feel like the experimental results on these pages are the most compelling part of the paper. Might it be possible for the authors to move it to earlier in the paper? (After maybe also introducing HUG and its concrete realizations earlier as well?)

**Typos**
* Figure 3 Caption: "verifyng"
* Appendix A General Settings: "gradient descend"


**Summary Of The Paper:**

This paper proposes an interesting new Hyperspherical Uniformity Gap (HUG) objective that explicitly minimizes within-class variation while maximizing between-class variation. An especially interesting innovation is introducing---instead of canonical linear classifiers---a "learnable proxy" that is minimized directly with variation measures in the loss.

**Summary Of The Review:**

The HUG objective is an interesting, well-motivated idea. The claims and experiments of this paper are promising. However, the paper, as is, suffers from some clarity issues. For example, there is no reproducible code and the exposition can be improved. Additionally, two very related prior works are missing.

I feel like the contributions of this paper are novel and worthy of publication. But the above-described issues make it only marginally so.

* I am willing to raise my score of this paper if the authors sufficiently address the above-described issues during the revision/discussion period.

---

> ### Author Response · Authors · 2022-11-19
> **Response to Reviewer k2o7 (Part 3)**
>
>
> **Q9: Page 5: "Nontrivial to optimize the original HUG..." At this point, the reader has not seen a concrete example of the HU(...) function yet. Thus, it is unclear why the original HUG is difficult to optimize. Even after seeing concrete examples in later papers, I am not sure which particular difficulties the authors are referring to.**
>
> A9: Sorry for the confusion. For the nontrivial part, we mostly mean computing the class feature mean is nontrivial, since it requires us to compute all the features in each class, which is inefficient for a large dataset. This is also why we propose the class proxy as a parametric center for each class (equivalent to a classifier). We will improve the clarity in the final revision.
>
>
> **Q10: Page 8-9: I feel like the experimental results on these pages are the most compelling part of the paper. Might it be possible for the authors to move it to earlier in the paper? (After maybe also introducing HUG and its concrete realizations earlier as well?)**
>
> A10: Great suggestion! We will restructure the paper and move the applications and experiments earlier in the final revision. Due to the limited amount of time, we have to leave the major update of the paper to the final revision.

---

> > ### Comment · Reviewer_k2o7 · 2022-11-19
> > **Response to Rebuttals**
> >
> > As in my original review, I find the *content* of this work novel and interesting. However, I feel like the *presentation* (even in the updated state) is, in many places, unpolished/unclear and discussions of prior works (still) need to be extended. The authors have promised to do this --- although the final results will be unclear until after the final version of the paper is uploaded, which will be after the review period of this paper.
> >
> > Nonetheless, the authors have (in their responses) provided promising and clarifying discussions in their responses to the reviews. Additionally, they have provided new figures, experiments, and a piece of illustrative code. Since the ICLR revision period is only two weeks, I feel like this constitutes a good faith effort by the authors to improve this paper. As such, I will give the authors the benefit-of-the-doubt that the presentation of this paper and discussion of prior works will be improved in the final version.
> >
> > **I have raised the score of this paper** with the following "footnote" to the AE, other reviewers, and OpenReview readers: This score reflects my belief of what the final version of the paper *could be* rather than what the current updated version *is*. I will hold the authors to their promise of improving the presentation and discussions in this paper in a later version.

---

> > > ### Author Response · Authors · 2022-11-21
> > > **Thanks!**
> > >
> > > Thanks for all the valuable suggestions for improving our paper! We are very glad that our rebuttal addressed the reviewer's concerns.

---

> ### Author Response · Authors · 2022-11-19
> **Response to Reviewer k2o7 (Part 2)**
>
>
>
> **Q4: Page 6-7: The implementations of MHE, MHS, and MGD all require the computation of all pairwise distance norms for the vectors passed to the associated HU(...) function. Computationally, this seems significantly slower than MSE or CE which do not require a quadratically-sized outer loop. Might the authors comment on this?**
>
> A4: Great question! Usually when the number of classes is not too large (say less than 1k), this will make neglectable difference in terms of training overhead (~10% more training time). But this small computational overhead will give us more benefits in generalization. When the number of classes is extremely large (say million-level number of classes), we will have to use mini-batch approximation for MHE to compute inter-class hyperspherical uniformity. Specifically, we can uniformly sample the classes and compute the MHE objective. This technique has been used in [Learning towards Minimum Hyperspherical Energy, NeurIPS 2018] for training deep face recognition networks (which typically involves >100k number of classes).
>
> While we agree with the reviewer that HUG may incur a bit more training overhead, we would like to emphasize that the inference (testing) time for both CE-trained networks and HUG-trained networks are exactly the same. Since the training time in modern machine learning is becoming less and less crucial, what really matters is the inference time. In terms of inference speed, HUG has no disadvantages at all.
>
>
> **Q5: On the same note: Might the authors discuss or include an experiment comparing the compute/CPU time required to train using HUG vs. using CE or MSE?**
>
> A5: Great suggestion! We compare the training time for CE and HUG under exactly the same settings. Compared to CE, we find that HUG-MHE yields 1.14x training time, HUG-MHS: yields 1.11x training time and HUG-MGD yields 1.23x training time. We will include all the training time comparison in the final version to address the reviewer’s concerns.
>
>
> **Q6: The authors discuss HUG extensively in pages 1-4, but HUG is not introduced until Page 5. As a result, for four entire pages, the reader is left to purely imagine what HUG (hypothetically) might be. This significantly weakens the authors' discussion. I strongly recommend introducing the formal definition of HUG earlier.**
>
> A6: Great suggestion! We agree with the reviewer that introducing HUG earlier can help readers better understand why GNC is well motivated and important. We will restructure the paper accordingly in the final version.
>
>
> **Q7: Page 2: Are Projection FDA and Data FDA instances of HUG with a particular choice of HU(...)? This is especially unclear at this point of this paper since HUG has not been introduced yet.**
>
> A7: Thanks for pointing this out. As the introduction section states, projection / data FDA essentially serve as a motivation for generalizing and decoupling NC to maximal inter-class separability and minimal intra-class variability. The FDA criterion has many degenerate solutions as discussed in the introduction section and is a necessary condition for the HUG objective. Therefore, in a high-level sense, FDA and HUG share the same ultimate goal, but technically, HUG places more constraints on the learned representations, since it rules out a lot of degenerate solutions.
>
>
> **Q8: Page 2: It seems there is an even bigger problem with Projection FDA and DataFDA as objectives: They require explicitly calculating the Sb and Sw of (at best) the minibatch and (at worse) the entire dataset. This is very memory and computationally expensive. The inverse will also be expensive. MSE and CE training do not require these matrix quantities and would be much more efficient. Might the authors comment on this?**
>
> A8: We agree with the reviewer that both projection FDA and data FDA are not scalable and highly expensive to optimize for large-scale datasets. This is exactly one of the reasons that we don’t choose to use projection / data FDA, but rather use them to draw inspirations. However, for MHE, MHS and MGD, they are much more efficient to compute and can also be optimized by mini-batch SGD. For example, on CIFAR-100 with ResNet-18, HUG with MHE is only slightly slower than CE loss (CE: 1x, HUG-MHE: 1.14x, HUG-MHS: 1.11x, HUG-MGD: 1.23x). Note that our implementation has not been optimized for better efficiency, so there is room for the training time of HUG to be further reduced.

---

> ### Author Response · Authors · 2022-11-19
> **Response to Reviewer k2o7 (Part 1)**
>
>
>
> We sincerely thank the reviewer for the positive and constructive comments. We are deeply appreciative for all the detailed suggestions as well as the recognition of our novelty. We take every raised question seriously and hope that our response can clarify your concerns. We will be more than happy to address any additional concerns.
>
>
> **Q1: The authors do not provide any reproducible code. Might is be possible for the authors to give short, self-contained code that (at least) shows the implementation of MHE, MHS, and MGD?**
>
> A1: We will definitely release the full version of our code to reproduce all the results in the paper. Now we give an example code for a clean implementation of HUG-MHE below:
>
> https://anonymous.4open.science/r/HUG-1355/HUG_loss.py
>
>
> **Q2: Lu and Steinerberger [39] also discusses the connection of hypersphere uniformity with neural collapse in their Section 1.4. Moreover, they discuss this relative to a "frame potential" along the same lines as to the energy minimization discussed by the authors of this paper's theory. I realize the authors already briefly acknowledge [39], but is done only in one passing sentence. Given the similarities in these two works, might the authors add a slightly longer discussion/comparison of these two works?**
>
> A2: Thanks for the great suggestion! [39] is indeed highly related and they have very nice proof for the hyperspherical uniformity for CE. This paper performs Taylor expansion on the loss functional, and the dominating terms show a strong connection to the frame potential. Minimizing this frame potential can actually be viewed as a special case of minimum hyperspherical energy. This result actually implies a strong link between CE loss and hyperspherical uniformity. Meanwhile, this work also recognizes the difficulty of showing that CE can converge to GNC when $d<C-1$ in the non-asymptotic case. We believe that this will become an important theoretical future work. In contrast to our work, [39] does not go beyond this connection and only connects to the inter-class hyperspherical uniformity without giving any usable loss function. Specifically, we design a framework for new loss functions (as an alternative to CE loss) by decoupling the GNC.
>
> We are sorry for not being able to give longer discussion in the submission version due to page limits, but we will definitely extend the above discussion in the final version.
>
>
> **Q3: The following ICML paper also examines neural collapse when the number of classes is larger than the ambient dimension: [Zhou, J., Li, X., Ding, T., You, C., Qu, Q. & Zhu, Z.. (2022). On the Optimization Landscape of Neural Collapse under MSE Loss: Global Optimality with Unconstrained Features]. Might the authors include additional discussions on how their GNC definition compares to the generalization of NC in the d<C-1 case given by Zhou et al. 2022?**
>
> A3: Huge thanks for providing this highly relevant and interesting paper! [Zhou et al.] indeed considers the case where $d<C-1$ and shows that the class mean matrix is the best d-rank approximation to the simplex ETF of $C$ classes. First of all, our hypothesis does not contradict with the conclusion drawn in [Zhou et al.], and more interestingly, along with [Zhou et al.], HUG induces a very interesting geometric explanation, which we believe will be of great interest to the community.
>
> We would like to show an interesting example for the abovementioned geometric explanation. Specifically, we consider a special case of $d=2$ and $C=4$. When $C=4$, the simplex ETF is given as the following figure. [Zhou et al.] shows that the global solution will be a $d$-rank approximation, which can be viewed as a 2D hyperplane in this case. Therefore, the global solution for $d=2$ and $C=4$ will lie on a hyperplane with a simplex ETF projected onto it. Our GNC shows another characterization of such a hyperplane. Combining these two results, we have an interesting observation that GNC and [Zhou et al.] yield an informative characterization of the global solution, as shown in the figure below. The complete geometric interpretation is given in the figure below.
>
> https://anonymous.4open.science/r/HUG_rebuttal-C4C5/rebuttal_fig.pdf

---

### Official Review · Reviewer_S3We · 2022-10-30

**Confidence:** 4
**Correctness:** 4
**Technical Novelty And Significance:** 3
**Empirical Novelty And Significance:** 3
**Recommendation:** 6

**Clarity, Quality, Novelty And Reproducibility:**

- Overall, the paper is well organized and most parts are very clear.
- Quality: this is well prepared as an academic work including theoretical proof and support experiments.
- The paper provides a generalized neural collapse definition and new loss functions, which appear to be new.
- In terms of reproducibility, the paper provides hyper-parameter settings for each experiment, but no code files or links are provided.

**Strength And Weaknesses:**

## Strength:
- Overall, the paper is well-written and the results are clearly presented.
- The authors slightly relax the definition of neural collapse to include the case where the feature dimension $d$ is smaller than the number of class $C$.
- The paper proposes a new class of objective functions based on the hyperspherical uniform gap (HUG), which captures the difference between inter-class and intra-class hyperspherical uniformity.
- The proposed loss improves upon CE in long-tailed recognition, continual learning, and adversarial robustness.

## Weakness:
- For the case the feature dimension $d$ is smaller than the number of class $C$, it seems that characterizing the geometry of maximally distant features on the unit sphere is still a challenging problem.  What is the minimum hyperspherical energy could be in this case? In other words, how could one tell if the features converge to GNC since the smallest hyperspherical energy is not zero?
- It was also observed in [A] that when the feature dimension $d$ is smaller than the number of class $C$, the features are still well separated via CE loss, but not for MSE loss. But there is no analysis for CE loss. Under the setting of unconstrained features, could the new losses or CE be proved to converge to a solution of the generalized NC for that case (i.e., feature dimension $d$ is much smaller than the number of classes (C))?


- I am curious whether normalization is necessary for the new objective, hyperspherical uniform gap. In other words, could the objective be extended to the case without normalization of the features?
- Typo: Caption of figure 3 c: is it about inter-class separability?
- Typo: Should -2<s<0 in Theorem 4?

[A] Zhou et al., On the Optimization Landscape of Neural Collapse under MSE Loss: Global Optimality with Unconstrained Features, ICML 2022.


**Summary Of The Paper:**

Inspired by the recently observed neural collapse (NC) phenomena in deep learning classifiers where the feature dimension is often large than the number of classes, this paper considers a more general format of neural collapse (GNC) to include the case where feature dimension $d$ is smaller than the number of class $C$. Then the authors decouple NC into two objectives: minimizing intra-class variability and maximizing inter-class separability on the hypersphere via the proposed hyperspherical uniformity. After decoupling these two objectives, the authors then propose a new objective, called hyperspherical uniform gap (HUG),  with three choices of measurement (MHE, MHS, and MGD) as examples. The proposed HUG-based loss shows some improvements in long-tailed recognition, continual learning, and adversarial robustness.

**Summary Of The Review:**

This paper introduces a more general NC definition and introduces a new objective (HUG) by decoupling the objectives into inter-class variability and inter-class separability. Some theoretical analyses are provided for both GNC and HUG. The proposed new loss functions can achieve better performance in long-tailed recognition, continual learning, and adversarial robustness. Overall, the results could be of interest to researchers working in these areas.

---

> ### Author Response · Authors · 2022-11-19
> **Response to Reviewer S3We (Part 2)**
>
> **Q2: It was also observed in [A] that when the feature dimension d is smaller than the number of classes C, the features are still well separated via CE loss, but not for MSE loss. But there is no analysis for CE loss. Under the setting of unconstrained features, could the new losses or CE be proved to converge to a solution of the generalized NC for that case (i.e., feature dimension  is much smaller than the number of classes (C))?**
>
> A2: Great question! For the new loss, we can show that GNC is indeed a maximizer. The proof is very straightforward, because maximal inter-class hyperspherical uniformity and minimal intra-class hyperspherical uniformity are independent and can simultaneously attain their optimum, which is exactly what we defined as GNC(1) and GNC(2).
>
> Since the HUG objective is non-convex, we have no guarantee that optimizing it with gradient descent leads to GNC. But this will be an interesting future work! However, as an encouraging sign, we empirically observe that GNC can be approximately attained, because both inter-class hyperspherical uniformity and intra-class hyperspherical uniformity (especially inter class uniformity) are close to optimum.
>
> For the CE loss, in the case of $d>C-1$, it can provably converge to GNC, since GNC will reduce to NC in this case. For the case of $d<C-1$, we don’t have a proof for the CE loss and this is why we call it a hypothesis. However, the empirical study in Section 2 of the main paper gives very positive evidence to support that the CE loss indeed converges to GNC when $d<C-1$. The only theoretical analysis for the $d<C-1$ case is the asymptotic case where $C\rightarrow \infty$, as given in Theorem 5 and the nice work of [39] (cited in the main paper).
>
> In order to verify how close the HUG objective is to the optimum, we conduct an experiment to optimize unconstrained features with HUG. We use exactly the same settings as the CIFAR-100 experiment: the same number of features, the same feature dimensions and the same initialization (random seed). For the unconstrained feature case, we have that the intra-class hyperspherical uniformity term (the second term in Eq.(6) of the main paper) approaches to 0, and the inter-class hyperspherical uniformity term (the first term in Eq.(6) of the main paper) becomes 0.45. The results below show that inter-class uniformity is well optimized by the HUG objective.
>
> ||Intra-class variability $\downarrow$ Cifar-10|inter-class uniformity $\downarrow$ Cifar-10|Intra-class variability $\downarrow$ Cifar-100| inter-class uniformity $\downarrow$ Cifar-100|
> |---|:---:|:---:|:---:|:---:|
> |unconstrained|0.018|0.450|0.02|0.495
> |HUG-MHE|0.290|0.450|0.295|0.495
>
>
> **Q3: I am curious whether normalization is necessary for the new objective, hyperspherical uniform gap. In other words, could the objective be extended to the case without normalization of the features?**
>
> A3: Excellent insight! We are also interested to see how HUG will work without the normalization. To answer the reviewer’s question, we have experimentally tested that the HUG objective can indeed be extended to the case without both feature and weight normalization. The extension is straightforward: we first replace the normalized weight vectors in the HUG-MHE objective with the original un-normalized weight vectors, and to avoid degenerate solutions, we then add a weight decay term with an additional hyperparameter controlling its strength. We empirically find that this un-normalized HUG objective can achieve similar performance (\~24% on CIFAR-100 with ResNet-18) as the normalized HUG objective. This is comparable to the original HUG objective (\~23.5%) and is better than CE (\~25%). We will add the full experimental results and related discussion of this unnormalized version to the final revision.
>
>
> **Q4: Typo: Caption of figure 3 c: is it about inter-class separability? Typo: Should -2<s<0 in Theorem 4?**
>
> A4: Thanks for pointing them out! We will fix them in revision.
>
>
> **Q5: In terms of reproducibility, the paper provides hyper-parameter settings for each experiment, but no code files or links are provided.**
>
> A5: Thanks for the comment. We will definitely release the code for reproducing the experimental results. In fact, this is also extremely easy to implement, see the following demo (using HUG-MHE to optimize unconstrained features) as an example.
>
> https://anonymous.4open.science/r/HUG-1355/HUG_loss.py

---

> ### Author Response · Authors · 2022-11-19
> **Response to Reviewer S3We (Part 1)**
>
> We sincerely thank the reviewer for the positive and constructive comments. We are super excited that the reviewer recognizes our novelty. We take every raised question seriously and hope that our response can clarify your concerns. We will be more than happy to address any additional concerns.
>
>
> **Q1: For the case the feature dimension d is smaller than the number of classes C, it seems that characterizing the geometry of maximally distant features on the unit sphere is still a challenging problem. What is the minimum hyperspherical energy could be in this case? In other words, how could one tell if the features converge to GNC since the smallest hyperspherical energy is not zero?**
>
> A1: Great question! We agree with the reviewer that the geometry of maximally distant features on the unit hypersphere is a difficult problem, and the exact value of the minimum hyperspherical energy in general cases is nontrivial to compute.
>
> To get an approximate value of the minimum hyperspherical energy, there are two strategies in general: (1) we can directly minimize the hyperspherical energy for unconstrained features with gradient descent (unconstrained features are freely trainable vectors, rather than being obtained from a neural encoder). Despite the potential of falling into a local minima, the result of this minimization can still serve as a good proxy to the minimum hyperspherical energy; (2) Since normalized zero-mean Gaussian random vectors are distributed uniformly over the unit hypersphere, we can run many trials for generating $C$ random vectors of $d$ dimensions, and then compute the average hyperspherical energy as a proxy to the minimum energy. Therefore, we can use these proxies to see whether we are close to reaching hyperspherical uniformity. This empirically serves as a good indicator to tell whether GNC is reached.
>
> As a side note, in the case of standard softmax cross-entropy loss, there is also no guaranteed way to tell whether NC or GNC is attained, because zero training loss does not necessarily lead to NC or GNC. A simple example is that, as long as the deeply learned features are linearly separable, then naively increasing all the classifier norm (i.e., making the classifier norm to an extremely large value) will lead to zero loss. This is because the softmax function will lead to one-hot coding for the largest confidence class once we natively increase either the feature norm or the classifier norm.

---

### Author Response · Authors · 2022-11-19
**Summary**

Dear Reviewers and ACs,

We sincerely thank all the reviewers and ACs for spending time on our submission. We are deeply appreciative for all the efforts from the reviewers and ACs put in improving our paper. We have responded to every raised concern and hope that our rebuttal can address them.

We are super excited that most of the reviewers recognize our novelty, and all reviewers find our problem significant. We notice that most concerns are about presentation and clarity. We are trying our best to improve the paper in such a short amount of time, and we will continue to revise / restructure our paper. **Every revision we promised to the reviewer will be made in the final version.** Besides that, we have conducted all the requested experiments and put them in the rebuttal.

If there is any more question or experiment that could help to address the reviewer’s concerns, please feel free to let us know. We will be more than happy to clarify any concern and conduct additional experiments.

The changed content for the updated paper is denoted as **cyan**. Note that the citation number has been changed because of some newly added references. To clarify, all the citation index we mentioned in the rebuttal refers to the original submission version. Moreover, if the current version does not reflect some of the changes, we will be making them in the final version.

For the final version, we will put some part of the paper to appendix in order to meet the page limit. For the current version, we leave them in the main paper for better readability.

Thanks again for all the effort and time.

Respectfully,

Authors

---

### Author Response · Authors · 2022-12-06
**Any additional question or concern?**

Dear Reviewers and ACs,

We sincerely thank all the reviewers and ACs for spending time on our submission.

We are wondering whether there are any remaining concerns or questions from the reviewers? We are more than happy to address them, both verbally and experimentally. Since there is only limited amount of time left, we will be deeply appreciative if the reviewers can let us know the concerns (if any) as soon as possible.

Thank you very much!

Respectfully,

Authors

---

### Decision · Program_Chairs · 2023-01-20

**Decision:**

Accept: poster

**Justification For Why Not Higher Score:**

One negative reviewer raised major concerns: 1) the correctness of hypothesis GNC(2), where the empirical evidence seems not strong yet and there is no proof for finite class case; 2) literature about discussing origins of theoretical results are not fully addressed. These concerns are indeed valuable and shortcomings of the current manuscript.

**Justification For Why Not Lower Score:**

Three reviewers are positive with this manuscript (two accept, one weakly accept), while one reviewer keeps to be negative after the discussions. So the decision follows the majority.

**Metareview: Summary, Strengths And Weaknesses:**

The manuscript considers the generalization of neural collapse (NC) phenomenon in deep learning classifiers from the original scenario that the feature dimension (d) is larger than the number of classes (C) to the scenario that the feature dimension is much smaller than the number of classes, i.e. $C-1>d$, where the simplex equiangular tight frame in NC fails. The generalization is summarized in GNC here. A key hypothesis in GNC (2) is raised that after full training, the class-means with respect to the global mean center minimize the hyperspherical energy defined as a function (Coulomb potential) of pairwise class mean distances, and are maximally distant on a hypersphere. This hypothesis is supported by the facts that (a) when $d>=C-1$, it reduces to a simplex Equiangular Tight Frame (ETF), which is observed by Papyan-Han-Donoho (PNAS, 2020) and later proved by Lu-Steinerberger (ACHA, 2022) and Fang-He-Long-Su (PNAS, 2022) for cross entropy loss, among others; (b) when $C->\infty$, Lu-Steinerberger (ACHA, 2022) showed that the class means converge to a uniformly distribution on the hypersphere, that minimizes the energy asymptotically. In addition, the authors empirically observed that the hyperspherical energy (which captures inter-class separability) decreases toward minima and intra-class variability measured by hyperspherical reverse-energy collapse toward 0 in the training process. Then the authors decouple GNC into two objectives: minimizing intra-class variability and maximizing inter-class separability on the hypersphere via the proposed hyperspherical uniformity. After decoupling these two objectives, the authors then propose a new objective, called hyperspherical uniform gap (HUG), with three choices of measurement (MHE, MHS, and MGD) as examples. The proposed HUG-based loss shows some improvements in long-tailed recognition, continual learning, and adversarial robustness.

The major concern from reviews is as follows: 1) the correctness of hypothesis GNC (2), where the empirical evidence is not extensively investigated and there is no proof for finite class number case; 2) related literature about connections and origins of theoretical results are not fully addressed. These concerns are indeed valuable for the authors' revision and readers' attention.

Despite of these shortcomings, the proposed GNC, especially hypothesis (2), is an interesting generalization of the neural collapse phenomenon for large number of classes ($d<C-1$), and the various connections observed in the manuscript are valuable and inspiring for future research. Thus the manuscript can be accepted had the authors made all the promised revisions in the final version, including the reproducible source codes and the necessary literature discussions even in the revised supplementary material.

**Note From Pc:**

if the above contains the word "oral" or "spotlight" please see: "oral" presentation means -> notable-top-5% and "spotlight" means -> notable-top-25%. As stated in our emails, we are disassociating presentation type from AC recommendations